# Visual processing of musical syntax and its relationship to sight-reading ability

Yeoeun Lim[1], Andrew Goldman[1,2]*

1 Cognitive Science Program, College of Arts and Sciences, Indiana University, Bloomington, Indiana, United States of America, 2 Department of Music Theory, Jacobs School of Music, Indiana University, Bloomington, Indiana, United States of America

* angoldm@iu.edu

## Abstract

Sight-reading—performing music by reading written notation without prior practice—is a fundamental skill for musicians, yet there is considerable variability in sight-reading ability that technical proficiency alone does not explain. Instead, cognitive mechanisms may explain differences in sight-reading expertise. Prediction of upcoming music is crucial for fluent sight-reading, and these predictions rely on knowledge of harmonic syntax—principles for ordering musical elements. Two research gaps remain: First, no study has directly examined how individual differences in sensitivity to harmonic syntax may relate to sight-reading proficiency. Second, while studies have shown that harmonic syntax can be processed through visual observation of musical actions without sound, it remains unclear whether similar sensitivity emerges during the reading of symbolic musical notation—a context more analogous to sight-reading, and parallel to word reading in language. To address these gaps, 20 skilled pianists completed three tasks: sight-reading, self-paced reading of chord progressions, and an audiation test. Using computational modeling to quantify harmonic predictability, we found that harmonically less predictable chords required significantly longer reading times, providing evidence that visual processing of symbolic music notation exhibits sensitivity to harmonic predictability. However, the degree of this sensitivity showed no significant correlation with sight-reading proficiency or audiation ability, although our sample size may have been too small to detect small interaction effects. Instead, accurate sight-readers showed faster chord reading times regardless of harmonic predictability, suggesting that general visual processing efficiency distinguishes proficient sight-readers. These findings extend prior evidence for visual harmonic processing to the domain of symbolic notation, suggesting that sensitivity to harmonic syntax may operate through written musical symbols without auditory input or motor imitation. Simultaneously, they suggest sight-reading may be a multidimensional skill where visual fluency may be more critical than sophisticated syntactic knowledge, suggesting the importance of developing visual processing efficiency for sight-reading expertise.

**Data availability statement:** All de-identified data and research materials are publicly available on the Open Science Framework (https://osf.io/gaz2t/). Our initially submitted revision was returned to us with a request for information about interview transcript data, but our data does not include any interview transcripts.

**Funding:** The author(s) received no specific funding for this work.

## Introduction

Sight-reading—the ability to read and perform musical notation without prior practice—is both a hallmark of musical talent and a fundamental skill for many professional musicians [1,2]. In contemporary practice, sight-reading remains crucial particularly for orchestral musicians, accompanists, and ensemble performers who must master vast repertoires within limited timeframes [3]. Music educators consistently rank it among the most important piano skills [4,5], and prestigious institutions (i.e., Associated Board of the Royal Schools of Music) incorporate sight-reading as a core component of comprehensive musicianship assessment [6].

Yet, exceptional piano technique does not necessarily translate into exceptional sight-reading ability, presenting an intriguing challenge for cognitive science. Professional pianists with comparable musical training and technical expertise exhibit individual differences in sight-reading proficiency [7–9]. Lehmann and Ericsson's [10] research systematically confirmed that solo performance specialists demonstrated poorer sight-reading abilities than accompaniment specialists (those who collaborate in small ensembles of musicians) despite similar years of training. This suggests that sight-reading and rehearsed performance follow separate developmental trajectories [11] and that cognitive mechanisms beyond technical proficiency are crucial for sight-reading achievement [12,13].

Understanding these individual differences requires examining the cognitive demands of real-time music reading and its execution. At its core, sight-reading involves simultaneous visual encoding, working memory storage, motor programming, and auditory monitoring [14,15]. Among these processes, the ability to predict upcoming musical content is particularly critical [16,17]. The importance of prediction is demonstrated through eye-hand span—the distance between the note currently being played (hand position) and the note being visually fixated (eye position) on the written score. Eye-tracking studies consistently show that skilled sight-readers maintain longer eye-hand spans than less skilled sight-readers, i.e., that they are reading music further ahead of what they are currently playing [18–27]. These longer eye-hand spans suggest that proficient sight-readers process more information ahead of time, creating an extended time cushion to decode the notation and prepare the necessary motor commands before execution. This advance preparation facilitates the formation of predictive representations of upcoming musical content. More intriguingly, Lim et al. [23] found that skilled sight-readers flexibly adjust their eye-hand span based on musical predictability—increasing it in low-entropy pieces to maximize anticipatory processing but reducing it in high-entropy pieces. Less proficient sight-readers maintained consistently short spans regardless of predictability. Similarly, Huovinen et al. [22] discovered an "early attraction" phenomenon where sight-readers' gaze is drawn to complex elements before reaching them, suggesting that predictive mechanisms operate both strategically and automatically.

The power of prediction sometimes overrides actual notation. Following Goldovsky's accidental observation that skilled sight-readers fail to notice printing errors as they automatically correct them from top-down predictions [9], Sloboda's [28] systematic experiment showed that participants detected errors well at

structurally important beginnings and endings of musical phrases but missed errors in predictable middle sections, indicating that structural knowledge operates as a predictive mechanism supplementing or replacing visual information. As Sloboda [29] explicitly expressed it, superior sight-reading is "the ability to decide on probable continuations within an idiom."

## Harmonic syntax as the foundation of musical prediction

What enables these predictions? The answer lies in musical syntax—"the principles governing the combination of discrete structural elements into sequences" [30]. While syntax exists across melodic, rhythmic, and metric dimensions [31], its regularities are most systematically manifested in Western tonal harmony.

Harmonic syntax governs how chords relate functionally and progress. At the basic level, it manifests as probabilistic tendencies for certain chords to lead to others. For example, in many styles of Western tonal music, the dominant chord (V) is often preceded by a predominant chord (such as IV) and followed by the tonic chord (I) with very high probability; accordingly, a harmonic progression like IV-V may generate a strong prediction of what will follow: I. These rules form hierarchical structures where local progressions serve specific functions within larger units like phrases [32,33]. Musicians acquire this syntactic system in part through statistical learning, implicitly internalizing probabilistic patterns through repeated exposure to specific musical styles [34,35]. This acquired knowledge automatically activates during real-time music processing to form expectations about upcoming events [36].

Evidence linking harmonic syntax to predictive mechanisms comes from multiple sources. Bharucha and Stoeckig's [37,38] harmonic priming studies showed that preceding chords facilitate processing of harmonically related subsequent chords, interpreted as pre-activation through syntactic predictions. At the neural level, the early right anterior negativity (ERAN) appears approximately 150–250ms after unexpected chords [39], functioning as a prediction error signal [40,41]. The P600 component—a positive ERP waveform observed in parietal-occipital regions approximately 500–800ms post-stimulus—reflects subsequent structural reanalysis attempts when predictions fail [42,43].

Thus, harmonic syntax may be particularly important in sight-reading contexts where most repertoire consists of Western tonal music. A variety of such pieces used in sight-reading education and assessment, including Baroque inventions (short pieces with two melodic lines played in counterpoint with each other) and Romantic character pieces (short pieces that are meant to evoke an idea, character, or scene), follow such principles of functional harmony. Several empirical studies show that harmonic context—though not strictly harmonic *syntax*—is a key source of prediction in sight-reading situations. In a study where skilled vocalists sight-sang original and manipulated versions of Bach chorales [16], researchers manipulated stimuli by altering melodies to disrupt pattern recognition or altering harmonies to disrupt tonal predictions. Results showed that participants' pitch errors significantly increased when harmonies were altered making prediction difficult, even though their own singing part remained completely unchanged. This demonstrates that performers predict upcoming notes using overall harmonic context rather than individual notes. Hadley et al. [44] captured through eye-movement tracking that skilled pianists showed immediate increases in eye regression and pupil dilation when encountering chords violating harmonic expectations during sight-reading. This suggests real-time processing burden increases when predictions fail, and that harmonic understanding is deeply integrated into the sight-reading process. Additionally, a qualitative study analyzing thought processes of advanced pianists sight-reading tonal, atonal, and ambiguously tonal music reported that pianists actively utilize harmonic knowledge like "intuitive grasp of harmonic flow" or "typical tonal schemas" to employ prediction strategies when sight-reading clearly tonal pieces [45].

## Harmonic syntax processing and sight-reading proficiency

If sight-reading depends on prediction based on harmonic *syntax*, would someone with greater sensitivity to harmonic syntax actually sight-read better? Surprisingly, no research has directly explored this question. While existing sight-reading studies have shown the importance of the somewhat broad concept of harmonic prediction, they have not specifically linked individuals' sensitivity to harmonic *syntax* with sight-reading achievement.

Several strands of indirect evidence suggest meaningful relationships may exist between harmonic syntax processing and sight-reading ability. First, performers with superior sight-reading skills tend to more effectively utilize harmonic context for prediction during sight-reading. Waters et al. [17] applied the harmonic priming paradigm to the visual modality by devising a task in which pianists viewed pairs of three-note chords presented sequentially in standard musical notation and judged the nature of chord combinations. Results showed skilled sight-readers demonstrated significant facilitation effects when processing tonally related chord pairs, while less-skilled sight-readers barely utilized these harmonic contextual advantages. More importantly, this harmonic prediction ability predicted sight-reading performance independently of basic pattern recognition abilities. Based on these results, the researchers suggested that "The better readers may be more sensitive to such [harmonic] regularities, and this may underlie their priming advantage over the poorer readers" (p. 140), suggesting indirect associations between harmonic syntax processing ability and sight-reading proficiency. More recently, Pomerleau-Turcotte et al. [46] quantitatively demonstrated that strategic use of such harmonic knowledge directly predicts sight-singing performance. Participants who used musical knowledge strategies like identifying scale degrees or chords during sight-singing tasks scored significantly higher in pitch and overall accuracy than those who did not. Notably, "using musical knowledge" emerged as the second strongest predictor of sight-singing performance after relying on automatic skills, systematically showing that understanding and utilizing harmonic structure is a key factor distinguishing individual sight-reading proficiency.

Second, general musical expertise enhances sensitivity to harmonic syntax violations. Neuroscientific evidence reveals that musicians show larger ERAN amplitudes than non-musicians when hearing chords violating harmonic rules [47]. These neural response differences appear from childhood—children with an average of 4 years and 9 months of musical training showed nearly twice the ERAN amplitude compared to untrained children [48]. This demonstrates that musical training forms more fine-grained and systematically organized neural representations of musical syntactic rules [49].

Taken together, these two strands of indirect evidence support the possibility of meaningful relationships between sight-reading expertise and harmonic *syntax* processing ability. Because sight-reading is a task requiring real-time grasp and prediction of musical structure, greater sensitivity to harmonic syntax—the ability to detect and utilize harmonic regularities—would provide crucial advantages in comprehending complex harmonic structures and predicting subsequent progressions within limited time. Nevertheless, these relationships remain as inference based on indirect evidence. No study to date has independently measured individuals' sensitivity to harmonic syntax and explored how it directly relates to actual sight-reading performance ability. This is one important research gap this study aims to fill.

### The modality gap: Can harmonic syntax be processed visually?

To establish this relationship, we must measure harmonic syntax processing in the sight-reading context. But here we face a fundamental problem: how to measure harmonic syntax processing in the context of sight-reading (specifically, sight-reading proficiency). This goes beyond being merely a methodological issue for sight-reading research—it is also an independent theoretical problem of the *modality gap* that harmonic syntax processing research has not addressed.

Most research on harmonic syntax processing has relied on the *auditory* modality. Findings that harmonic syntax violations elicit characteristic brainwave responses like ERAN, N5 (negativity peaking around 500ms after presenting unexpected/irregular chords, analogous to N400 in language), and P600 were all made through auditory stimuli (i.e., tasks where chords are heard) [39,43,50–54]. However, sight-reading essentially begins with processing visual information, i.e., musical notation. Skilled sight-readers grasp harmonic structure and predict progressions through notation before fingers touch keys, before sounds are produced. That is, at the core of the complex act of sight-reading, the process of pure visual syntax processing—grasping syntactic relationships through visual symbols alone *before* hearing sounds—necessarily comes first. Yet during actual sight-reading performance, such visual processing is intertwined in real time with auditory and motor processing, making it difficult to independently measure pure visual syntax processing ability. At the moment performers look at notation, they already hear sounds from previously played notes, and fingers are moving toward the next notes.

Interestingly, while musicians may engage in harmonic analysis from notation alone (e.g., in a music theory classroom examining a musical score by eye), it is yet uncertain whether such visual processing exhibits similar sensitivity to harmonic predictability observed in auditory paradigms. In studies using auditory stimuli, listeners show characteristic behavioral responses such as facilitated processing of harmonically related chords [37,38] and neural responses such as ERAN and P600 to harmonically unexpected events [39,43], suggesting that expectation-based mechanisms operate during real-time harmonic processing. Whether similar mechanisms operate when harmonic information is acquired visually remains an open question. If visual processing of notation engages the same predictive mechanisms as auditory processing—showing graded sensitivity to harmonic predictability—this would suggest that harmonic syntax is represented at an abstract level that might transcend specific sensory modalities. Alternatively, if harmonic expectations are fundamentally tied to auditory experience, visual notation reading might engage different cognitive processes altogether.

This question has been extensively explored in language. Both auditory and visual modalities show real-time syntax processing. Grammatical violations heard in speech elicit ELAN (100–200ms) and P600 responses [55,56]; see also Friederici [57] for detailed discussion of auditory sentence processing), while reading shows longer reading times ([58–60]; see also Yoshida [61] for a review) or longer fixations (Frazier and Rayner [62,63]; see also Rayner [64] for additional review) at syntactically difficult points as well as similar P600 responses to visual grammatical errors [65–67]. Direct comparisons confirm that syntax processing transcends modalities—P600 appears for both visual and auditory grammatical violations with similar timing and distribution [65,68]. fMRI studies show that language networks including Broca's area activate regardless of input modality [69,70].

This balanced approach to syntax processing modalities in the language domain contrasts with that in the music domain. Almost all research on harmonic syntax processing in music has relied entirely on auditory stimuli. Of course, there have been a few pioneering studies utilizing visual music notation. However, these studies have not yet provided clear answers to the question of pure visual syntax processing due to their respective limitations. For example, the previously mentioned Waters et al. [17] study showed harmonic priming effects with notation alone, but this focused on associative relationships between individual chords rather than hierarchical syntax processing. Also, Kragness and Trainor's [71] study finding "boundary dwelling" phenomena where participants linger longer at phrase boundaries with harmonic cadences in self-paced reading tasks suggests visually acquired harmonic information influences phrase segmentation, but this is closer to detecting local boundary signals rather than hierarchical processing of complex syntactic structures. Schön and Besson [72] and Shin and Fujioka [73] presented visual and auditory stimuli simultaneously, preventing isolation of purely visual processing; Gunter et al. [74] used the diatonic violations within familiar melodies where memory could contaminate results. The study that most directly explored syntax processing through visual modality is perhaps an eye-tracking study conducted by Ahken et al. [75]. They found participants showed similar gaze patterns (increased fixation time, etc.) when reading linguistic and musical syntax violations, suggesting possible common processing between domains. However, their musical task was sight-reading—simultaneously looking at notation while performing. This limitation meant pure visual processing could not be separated from motor planning and execution, and consequent auditory feedback effects.

Beyond studies using musical notation, researchers have demonstrated that harmonic syntax can be processed in the absence of auditory input through visual observation of musical actions. Sammler et al. [76] showed that pianists watching silent videos of hands playing chord progressions exhibited ERP signatures sensitive to harmonic violations, and Bianco et al. [77] revealed distinct fronto-parietal networks for processing harmonic structure during silent imitation of photographed hand movements. These studies provide evidence suggesting that harmonic syntax processing is not strictly bound to the auditory modality but can also be processed through the visuomotor system via action observation and motor simulation. However, these paradigms differ from notation-based processing in potentially important ways. Both studies employed visual depictions of musical actions—videos or photographs of hands playing chords—rather than symbolic musical notation. Moreover, in these visual/action conditions, participants were required to motorically imitate the

observed movements, making it difficult to isolate visual processing from motor planning and execution. This leaves open the question of whether harmonic syntax can be extracted from symbolic notation alone—the abstract system through which musicians typically encounter musical structure in sight-reading contexts—without the support of visuomotor simulation. This question represents both a theoretical issue regarding the potential abstractness of harmonic representations and a practical consideration for understanding the relationship between notation-based harmonic processing and sight-reading proficiency.

## The current study

Based on these interrelated gaps, the present study addresses two main questions: (1) Can harmonic syntax be processed through visual notation alone? (2) If such visual processing is possible, how does this ability relate to sight-reading proficiency?

We designed a behavioral experiment combining three tasks with skilled pianists. For pure visual syntax processing, we adapted the self-paced reading paradigm from psycholinguistics, where reading time indicates cognitive processing effort. Unlike previous binary "violation" paradigms (i.e., typical vs. violated chords), we use natural harmonic progressions taken from actual musical repertoire while quantifying the predictability of each chord as continuous values through computational modeling. This reflects the reality that harmonic expectations encountered in actual music, particularly in sight-reading situations, exist on a probabilistic spectrum rather than simple "right or wrong." We believe this approach will also enhance the ecological validity of the study.

It is important to clarify what we mean by visual syntax processing in this context. We conceptualize this as the ability to extract harmonic relationships from musical notation without auditory input, measured through sensitivity to harmonic predictability. Specifically, we assess this sensitivity through the degree to which reading times increase according to the harmonic predictability of the chords in the self-paced reading task. Such sensitivity—reflected in longer reading times for less predictable chords—would suggest that readers are engaging with harmonic relationships during silent music reading, as their reading behavior appears to be modulated by the statistical properties of the harmonic sequence.

For the relationship between visual syntax processing and sight-reading, participants sight-read both tonal and atonal pieces. Tonal music follows the same syntactic regularities use to model the stimuli in our self-paced reading task; atonal music avoids these regularities and is harmonically much less predictable. If visual syntax processing matters for sight-reading, performance on tonal pieces should correlate more strongly with sensitivity to syntactic typicality than performance on atonal pieces. Through this comparison, we expect to reveal how harmonic syntax processing ability independently contributes to sight-reading.

Finally, an audiation task explores whether visual syntax processing relies on abstract symbolic prediction or internal auditory imagery, as [notational] audiation—"hearing music from notation when the sound is not physically present" [78]—may mediate between visual processing and sight-reading performance [79].

We hypothesize that: (1) Harmonic syntax can be processed through pure visual information alone, without auditory input. Accordingly, we expect chords with low harmonic predictability to require significantly longer reading times than highly predictable chords in the self-paced reading task; (2) Sensitivity in visual syntax processing will correlate positively with sight-reading proficiency, and this relationship will be specific to tonal sight-reading. Specifically, participants who show larger reading time differences between less predictable and predictable chords (indicating greater sensitivity to harmonic syntax) will demonstrate better sight-reading performance—but this correlation will emerge only when sight-reading tonal pieces, and will be absent or weaker for atonal pieces; (3) Audiation ability will moderate this relationship, with the strongest effects for participants high in both sight-reading proficiency and audiation.

## Materials and methods

### Ethics statement

This study was approved by the Indiana University Institutional Review Board (Protocol #25242) under exempt review procedures. Participants read a Study Information Sheet and confirmed their agreement to participate before beginning the

experiment. Written consent was not required for this minimal-risk study as determined by the IRB. All procedures were conducted in accordance with the ethical guidelines of the IRB.

## Participants

A total of 23 pianists were recruited for this study. Data from three participants were excluded due to failure to follow task instructions (see S1 File for detailed exclusion criteria), resulting in a final sample of 20 participants (15 females, 5 males; mean age = 24.9 years, $SD$ = 2.34, range: 22–31). Our sample size ($N$ = 20) was determined based on established precedents in sight-reading research. This sample size is comparable to similar studies investigating cognitive mechanisms of piano sight-reading (e.g., Aiba and Matsui [80], $N$ = 11; Arthur et al. [81], $N$ = 21; Cara [18,19], $N$ = 22; Fine et al. [16], $N$ = 22; Kim et al. [45], $N$ = 12; Lehmann and Ericsson [10], $N$ = 16; Waters et al. [17], $N$ = 18; Zhukov et al. [82], $N$ = 25), with eye-tracking studies—which similarly require individual testing—successfully detecting effects with even smaller samples (e.g., Furneaux and Land [20], $N$ = 8; Rosemann et al. [83], $N$ = 9; Wurtz et al. [27], $N$ = 7; Zhukov et al. [3], $N$ = 6). Participants were recruited from our university and received $30 compensation. Inclusion criteria required current enrollment in or graduation from university-level piano programs, normal or corrected-to-normal vision and hearing, and no performance-impairing injuries. Data collection occurred in 2025.

Table 1 presents participant characteristics. The sample consisted primarily of graduate students (90%) who began piano at an average age of 5.1 years with nearly 10 years of formal training. Self-reported sight-reading confidence was moderate overall ($M$ = 4.80, $SD$ = 1.58 on a 7-point scale), with higher confidence for tonal ($M$ = 5.55) versus atonal music ($M$ = 2.95). Most participants engaged in sight-reading at least weekly, primarily in accompaniment contexts.

## Sight-reading task

**Materials and stimulus design.** For the sight-reading materials, we used the Associated Board of the Royal Schools of Music (ABRSM) Piano Specimen Sight-Reading Tests Grade 8, which is a widely recognized and standardized piano sight-reading assessment tool [for sight-reading studies using the ABRSM test materials, see 6, 20, 84, 85]. A total of six pieces were used: three tonal pieces selected from the ABRSM collection (Nos. 4, 13, and 15) and three atonal counterparts created by adapting other pieces from the same collection (Nos. 14, 6, and 10). Atonal versions preserved rhythm, meter, and texture while removing functional harmony through systematic pitch alterations—shifting selected pitches by semitones or whole tones to eliminate tonal relationships while maintaining playability (see S1 File for detailed atonalization procedures). The resulting sight-reading materials encompassed various keys and meters. Appropriate playing tempo were set for each piece based on its character and technical demands, with a metronome ensuring consistent timing across all participants.

The evaluation of sight-reading performance in this study focused solely on pitch and rhythm accuracy (see Data Analysis). Musical expression was intentionally excluded from the analysis because accuracy constitutes the fundamental substrate of proficient sight-reading and focusing on it allows for a more objective, interpretable measure of performance, free from the confounding variable of individual artistic interpretation. For that reason, all original markings related to tempo changes, such as *ritardandos* (gradual tempo decreases) or fermatas (held notes), were removed from the scores to ensure consistent analysis conditions across all participants. However, we preserved all other original markings, like dynamics (indicating volume) and articulations (instructions to play a note sharply, smoothly, etc.), to maintain the naturalistic sight-reading context for the participants, though any expressive characteristics of the performances were not included in the evaluation of sight-reading performance. Each participant performed three tonal and three atonal pieces, and the assignment of pieces was counterbalanced across participants to prevent ordering effects.

**Objective validation of stimuli.** To objectively verify that our atonal pieces exhibited lower predictability than the tonal ones, we analyzed their melodic predictability (i.e., how expected or surprising each note is given the preceding context) using the Information Dynamics of Music (IDyOM) [34]. IDyOM is a variable-order Markov model that learns sequential

**Table 1. Participant characteristics (*N*=20).**

| Demographic | N (%) |
|---|---|
| Age, years, mean±SD | 24.9±2.34 |
| Gender | |
| Female | 15 (75) |
| Male | 5 (25) |
| Musical expertise | |
| Graduate level | 18 (90) |
| Undergraduate level | 2 (10) |
| Gold-MSI Perceptual Abilities subscale, percentile (Range) | 50 (8–86) |
| Gold-MSI Musical Training subscale, percentile (Range) | 89 (74–99) |
| Age started piano, years, mean±SD | 5.12±1.15 |
| University-level training, years, mean±SD | 6.78±1.53 |
| Sight-reading frequency | |
| Daily to weekly | 15 (75) |
| Monthly or less | 5 (25) |
| Self-reported sight-reading confidence (1–7), mean±SD | |
| Tonal | 5.55±1.19 |
| Atonal | 2.95±1.67 |
| Overall | 4.80±1.58 |

Abbreviations: Gold-MSI: Goldsmiths musical sophistication index; M: mean; SD: standard deviation.

Sight-reading frequency breakdown: every day (5%), 5–6 times/week (5%), 3–4 times/week (20%), 1–2 times/week (45%), monthly (15%), yearly or rarely (10%).

statistical regularities from musical corpora and generates probabilistic predictions for musical events. The model acquires internal representations of statistical regularities from exposure to a musical corpus and uses these learned distributions to estimate the predictability of events in novel sequences. This predictability is expressed as Information Content (IC), defined as the negative log probability of an event given its preceding context (IC = $-log_2 P(event \mid context)$); higher IC values correspond to less expected events according to the model's learned distributions.

We employed the "both+" model configuration, which is the default setting in IDyOM. This configuration combines two learning components: a long-term model (LTM) that learns statistical regularities from a pretraining corpus, representing knowledge that might approximate a listener's accumulated musical experience, and a short-term model (STM) that begins empty and learns incrementally within each piece, capturing local, piece-specific patterns such as recurring motifs. The "+" indicates that the LTM additionally receives incremental updates from the test sequence during processing, which is intended to simulate how listeners may continuously refine their expectations based on incoming musical information [34,86]. The LTM was pretrained on two external corpora: the Bach 371 Chorales, a collection of four-part chorales available from the Center for Computer Assisted Research in the Humanities (CCARH) Kern Scores collection, and the Kostka-Payne (KP) corpus [87], which consists of 46 excerpts from common-practice repertoire. These corpora were selected to provide the model with broad exposure to Western tonal conventions.

Rather than manually specifying source viewpoints, we used IDyOM's automatic viewpoint selection procedure, which employs a hill-climbing algorithm to identify the combination of viewpoints that minimizes mean IC for the target viewpoint. Viewpoints in IDyOM are representations of musical features; they can be basic (direct event attributes such as pitch), derived (computed from basic attributes, such as pitch interval), or linked (combinations of multiple viewpoints). The target

viewpoint in our melodic IC analysis was chromatic pitch (*cpitch*), and the selection procedure identified the following source viewpoints: *cpintfref* (chromatic pitch interval from the referent, representing scale degree), *cpint* (chromatic pitch interval from the preceding note), *(cpitch cpintfref)* (a linked viewpoint combining absolute pitch with scale degree), and *(cpint contour)* (a linked viewpoint combining interval with melodic contour direction, where −1 = descending, 0 = unison, 1 = ascending). This combination captures both absolute and relative pitch information, as well as interactions between intervallic and contour-based melodic features.

To prevent the model from being tested on data it had seen during training, we also employed leave-one-out cross-validation (k = 6). Under this procedure, when computing IC values for any given piece, that piece itself was excluded from the LTM training data; here, the model was trained only on the remaining five sight-reading pieces together with the external pretraining corpora. The melodic IC values of the sight-reading materials, including the raw IDyOM output files and the IDyOM Lisp commands used to generate these output files are available in the Open Science Framework (OSF) repository for this project at https://osf.io/gaz2t/.

The analysis, which calculated the melodic IC for each note, confirmed our manipulation was successful. Due to non-normal distributions in the melodic IC of both sight-reading conditions (Shapiro-Wilk tests, both $ps < .001$), we conducted a Mann-Whitney U test, which revealed that the atonal pieces ($n = 265$) had significantly higher melodic IC ($Mdn = 3.42$, $IQR = 2.66$) than the tonal pieces ($n = 279$; $Mdn = 2.61$, $IQR = 3.29$), $U = 43520$, $p < .001$, $r = −.18$. This result provides quantitative evidence that our atonal pieces were, on average, less predictable than the tonal ones according to the model.

### Self-paced reading task

To examine whether harmonic syntax can be processed through visual notation alone, we adapted the self-paced reading paradigm from psycholinguistics. This paradigm has been widely used to examine on-line processing of linguistic structure, with an assumption—sometimes called the "immediacy assumption" [88]—that readers tend to process each element as it is encountered and that the time spent on each element may reflect the cognitive demands of that processing. Support for this assumption comes from research comparing self-paced reading with eye-tracking, which has shown generally similar word-level effects across the two paradigms [89], as well as from information-theoretic research demonstrating relationships between surprisal and reading times across diverse constructions [90–92]. By applying this paradigm to musical chord reading, we aimed to assess whether reading times would vary systematically with harmonic predictability.

**Stimuli and experimental design.** The stimuli consisted of 46 chord progressions, with four notes per chord ("four-part"), sourced from the previously mentioned KP corpus. The process of harmonic reduction involved several key steps. Each musical excerpt from the corpus was reduced to a standardized four-part texture following principles similar to Bach-style chorale writing. In cases where the original excerpt contained chords with more than four parts, we removed the extra notes following music-theoretical principles (by preserving functional chord tones, e.g., root, third, seventh, and eliminating non-essential doublings while maintaining harmonic function). Voicings were adjusted for clarity, ensuring that harmonic structures were legible for real-time visual reading, and chord inversions were retained as much as possible to preserve original harmonic syntax. Each progression consisted of the first eight chords extracted from the original musical excerpt. To ensure visual consistency across trials, chord durations were standardized to whole notes, and key signatures were retained to provide an appropriate tonal framework for participants. As a result, some progressions may not conclude with a complete cadential resolution; however, this was acceptable for our purposes, as the primary goal was to examine reading time variations across chords of different harmonic predictability rather than to assess responses to phrase-final closure. Fig 1 shows an example of the four-part chord progressions used in the task. All chord progression stimuli, including MIDI and image files, are available in the OSF repository (https://osf.io/gaz2t/).

To ensure participants carefully engaged with the chord progressions rather than clicking through them mechanically, each trial concluded with a judgment question (e.g., "Was this chord included in the progression you just read?"). These

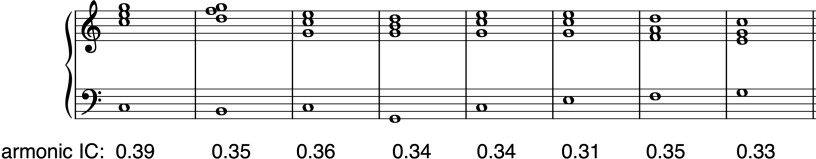

Harmonic IC: 0.39 0.35 0.36 0.34 0.34 0.31 0.35 0.33

**Fig 1. Example stimulus from the self-paced reading task.** *Note*. This example is taken from the experimental stimulus set (Stimulus ID: mzt.ekn. II). Numbers below each chord indicate the harmonic information content (IC) values calculated by the Information Dynamics of Music (IDyOM). Higher IC values indicate lower conditional probability given the preceding context—that is, events that are less expected based on the statistical regularities learned by the model. This example illustrates a pattern commonly observed in our stimulus set: harmonic IC values tend to decrease toward the end of progressions. This pattern likely reflects the conventional cadential structure of our stimuli, in which accumulating context combined with stylistically predictable harmonic patterns may constrain model predictions for later events, making them more predictable (lower IC values). We note that this is not a universal property of the model; unexpected events can receive high IC values regardless of position.

questions were created by randomly selecting one chord from each progression, which was either presented unchanged (50% of trials) or modified by changing its quality (e.g., F major to F minor; 50% of trials). The task was implemented and presented using PsychoPy software.

**Harmonic predictability analysis.** This study adopted a probabilistic approach to harmonic syntax. Within this framework, the predictability of a given chord reflects its syntactic typicality. Our primary goal was therefore to quantify this harmonic predictability for each chord in the self-paced reading stimuli. To this end, we again used IDyOM. Our central assumption is that harmonic IC, calculated by the model, can serve as a quantitative proxy for the cognitive demands of processing harmonic syntax.

In contrast to the melodic IC analysis of the sight-reading materials, this harmonic analysis employed IDyOM's harmony modeling functionality, which processes vertical sonorities (chords) as unified sequential events rather than individual pitches. As documented in the official IDyOM tutorial (https://github.com/mtpearce/idyom-tutorial), this is achieved by specifying the *:texture:harmony* parameter, which instructs the model to represent musical input as a sequence of vertical slices where each chord constitutes a single event.

The target viewpoint used in our harmonic IC analysis was *h-cpitch*, which represents each chord as the complete set of simultaneously sounding pitches. Following the examples in the tutorial, we used three source viewpoints that capture complementary aspects of harmonic structure: *pc-set* (the unordered pitch-class set present in the chord, abstracting away voicing and octave placement; e.g., a C major triad in any voicing yields the set {0, 4, 7}), *pc-set-rel-bass* (the pitch-class content expressed relative to the bass note, capturing information about chord inversion), and *root-sd* (the pitch-class of the chord root relative to the tonic of the prevailing key, representing harmonic function; e.g., 0 semitones = tonic chord, 7 semitones = dominant chord). Each source viewpoint generates its own probability distribution for the target. IDyOM then combines these distributions by computing a weighted geometric mean, where the weight assigned to each viewpoint is inversely related to its entropy—that is, viewpoints with more confident (less uncertain) predictions contribute more strongly to the final estimate [86]. We note that automatic viewpoint selection does not appear to be supported for harmonic texture in the current IDyOM implementation; therefore, viewpoints were selected based on the tutorial documentation.

We used the same "both+" model configuration, which is the default, and pretraining corpora (Bach 371 Chorales and KP corpus) as in the preceding analysis of melodic IC of the sight-reading pieces. We also employed leave-one-out cross-validation (k = 46), ensuring that when computing IC values for any given chord progression, that progression was excluded from the LTM training data; the model was trained only on the remaining 45 progressions together with the external pretraining corpora. The distribution of these harmonic IC values (Fig 2) showed variability in harmonic predictability across our self-paced reading stimuli to test the hypothesized relationship between harmonic IC and reading times. The

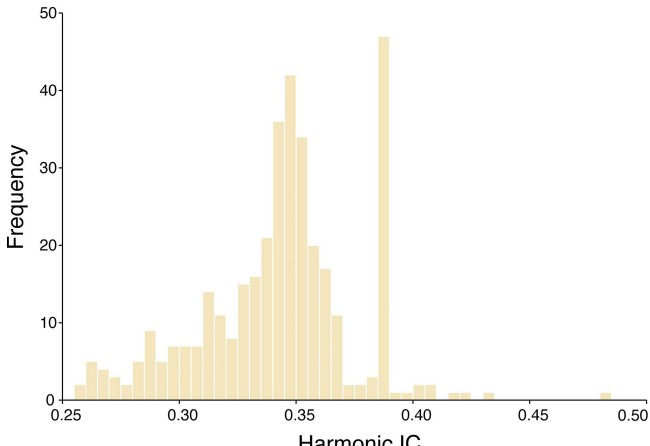

**Fig 2. Distribution of harmonic information content (IC) across all stimuli in the self-paced reading task.** *Note. N* = 368 chords (46 progressions × 8 chords/progression). The prominent spike (0.39) represents all first chords, which received identical harmonic IC values from the Information Dynamics of Music (IDyOM) as they lack preceding context. These first-chord data were excluded from subsequent analyses to accommodate temporal dependencies in the statistical modeling. The remaining distribution shows variability in harmonic predictability across the stimuli.

resulting harmonic IC values were used as a continuous predictor in subsequent analyses of reading times. The complete harmonic IC values for each chord and raw IDyOM output files are available in the OSF repository (https://osf.io/gaz2t/).

## Audiation task

To assess participants' audiation ability, we used the Notation-Evoked Sound Imagery test (NESI; Wolf et al. [93]). In this test, participants view a melody in notation and imagine how it sounds, then hear a melody and determine if it harmonically matches their imagination. The test consists of 12 questions requiring binary (yes/no) responses.

Each item in the NESI test consists of three interconnected melodies: (a) a melody, (b) a melody with added "figural" variations (essentially, additional notes that ornament the melody) that harmonically matched the original, and (c) a lure, which was a harmonically mismatching but plausible variation. During the test, participants were first presented with either a variation or a lure in musical notation and were instructed to imagine this melody as clearly as possible. After indicating they had completed this mental imagery process, the notation disappeared, and they heard the corresponding theme. Participants then judged whether the imagined melody (from notation) harmonically matched the heard theme.

While the original NESI test is available online (https://www.thinkinginmusic.com/?setLang=en), the present study re-implemented the test in PsychoPy to facilitate more detailed data collection and analysis. The implementation maintained the same structure of the original, consisting of both a training session and the main test session.

The final score for this task was calculated from the total number of correct responses across all 12 items, normalized to a 100-point scale to facilitate interpretation and comparison with other measures.

## Questionnaire

Participants completed a questionnaire designed to gather demographic information and details about their musical background and experience. The questionnaire included items assessing demographics, specific sight-reading experiences, and relevant subscales from the Goldsmiths Musical Sophistication Index (Gold-MSI; Müllensiefen et al. [94]), namely Perceptual Abilities and Musical Training. The data collected from this questionnaire were used to provide the detailed characterization of the sample presented in the Participants section. The full questionnaire is available in the S1 File.

## Procedure

Upon arriving at the laboratory, participants first read a Study Information Sheet and confirmed their agreement to participate before beginning the experiment. The experimental session consisted of four tasks presented in a fixed order: the sight-reading task, the self-paced reading task, the audiation task, and a questionnaire.

The session began with the sight-reading task (approximately 20 minutes). For this task, participants were seated at a Casio CDP-S360 digital piano, and their performances were recorded as MIDI files using Logic Pro X (version 10.8.1). The six pieces (three tonal and three atonal) were presented in a random, counterbalanced order across participants. Before beginning each piece, participants were first shown only the key and time signatures for approximately 10 seconds. Immediately after this preview, the full score was presented. To control for tempo differences (and thus enabling fair inter-subject comparisons), participants then heard four measures of a metronome beat and were instructed to maintain this tempo throughout their performance. The metronome continued throughout the entire performance. They were instructed to "play as accurately and fluently as possible," to "prioritize playing the correct notes fluently," and to continue playing without stopping if they made a mistake. All six pieces were performed consecutively with a short break of less than one minute between each.

Next, participants completed the self-paced reading task (approximately 30 minutes). The task consisted of 46 trials. To familiarize participants with the self-paced reading paradigm, the main experiment was preceded by a four-trial practice session. In each trial, participants viewed a chord progression using a moving-window presentation: the key signature with clefs for each staff appeared on the left side of the screen, and each spacebar press revealed the next chord at the following position to the right while the previous chord disappeared, so that only one chord was visible at a time (Fig 3). The key signature remained visible throughout each trial. Participants were instructed to read each chord carefully at a natural pace, ensuring they understood each chord before proceeding, and to avoid clicking through too quickly or mechanically. Reading time (RT)—the duration between successive spacebar presses—was recorded for each chord. After viewing each full progression, participants answered a yes/no judgment question about whether a probe chord had appeared in the progression. This task was conducted in a soundproof room, and participants were instructed to read the notation silently.

The audiation task (approximately 20 minutes) was administered through PsychoPy as well. The task began with a practice session that included detailed instructions and two practice trials with feedback in case of wrong answers. These instructions were followed by the main experiment where the 12 NESI questions were presented in a counterbalanced order across participants. As this task required silent music reading and audio listening, it was also conducted in the soundproof room with participants wearing noise cancelling headphones (Bose QuietComfort Headphones).

After completing the sight-reading, self-paced reading, and audiation tasks, the questionnaire (10 minutes) was administered in paper format, with participants writing their answers directly on the form. The entire experimental session lasted approximately 80 minutes, and participants received their compensation upon completion.

## Data analysis

**Sight-reading performance.** Our evaluation of sight-reading performance centered on accuracy—pitch and rhythm—deliberately excluding expressive elements to establish objective, comparable measures across participants. Accuracy constitutes the foundational substrate of proficient sight-reading upon which expressive performance is built.

All MIDI performance data underwent initial alignment to corresponding reference scores using the algorithm proposed by Nakamura et al. [95]. This probabilistic model finds optimal correspondence between performed notes and the position of their corresponding symbols in the score, accounting for performance errors and timing variations. The algorithm employs dynamic programming to match notes based on both pitch and timing information, outputting detailed correspondence files where each performed note is matched to its score position and classified by error type: correct notes,

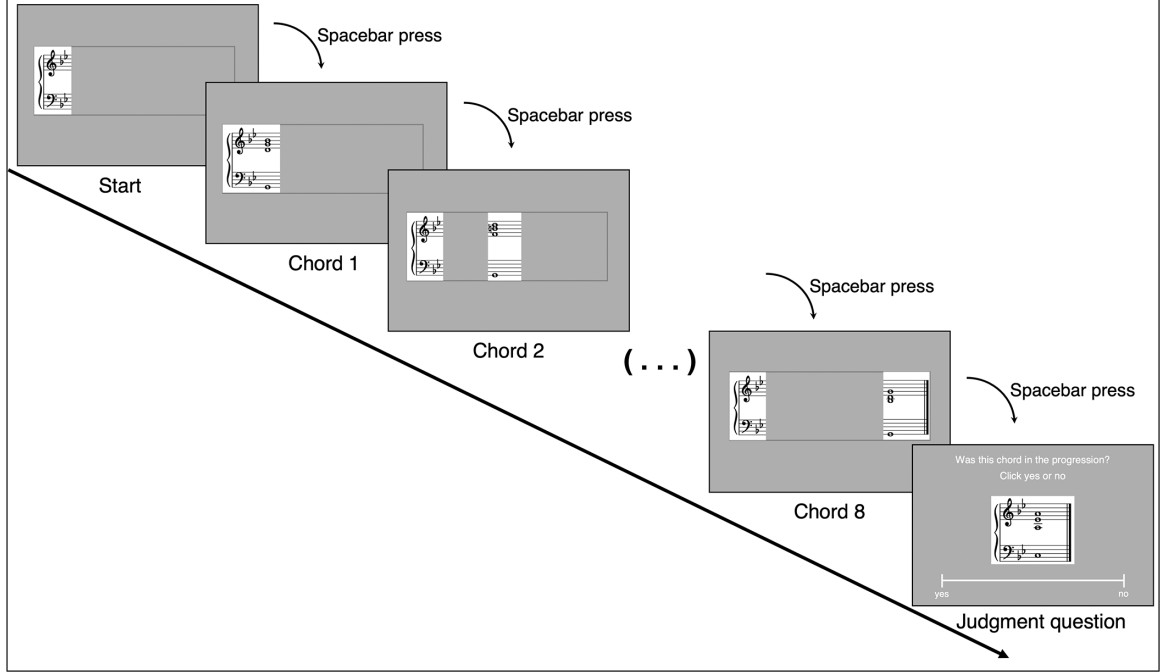

**Fig 3. Schematic illustration of the self-paced reading task.** *Note.* Participants viewed chord progressions one chord at a time using a moving-window presentation method. At trial onset, only the key signature was displayed. Each spacebar press revealed the next chord at a new position to the right while the previous chord disappeared. The key signature remained visible throughout. Reading time (RT) for each chord was measured as the duration between successive spacebar presses. After viewing the complete progression, participants judged whether a probe chord had appeared in the sequence.

pitch errors (correct timing but wrong pitch), extra notes (not in the score), or missing notes (in the score but not played). Additionally, the alignment provides precise timing information for each matched note.

Performance onset times were normalized to correct for variations in when participants began playing relative to MIDI recording start. We identified each participant's first performed note (as determined by alignment) and calculated its offset from zero, then subtracted this offset from all note timings throughout that performance, effectively aligning all performances to a common temporal framework where the first note begins at $t = 0$.

We calculated three primary accuracy metrics: pitch error rate, rhythm precision (beat-normalized mean absolute deviation; bn-MAD), and rhythm consistency (coefficient of variation; *CV*). Pitch error rate quantified overall note accuracy as the proportion of errors relative to total score notes:

$$Pitch\ Error\ Rate = \frac{\#\ pitch\ errors + \#\ missing\ notes + \#\ extra\ notes}{\#\ total\ notes\ in\ score} \tag{1}$$

This metric ranges from 0 to 1, where lower values indicate better—more accurate—performance. By including all three error types, this measure comprehensively assesses how faithfully participants reproduced the written score.

Rhythm accuracy was assessed using two complementary metrics capturing different temporal aspects: how precisely notes align with their intended timing (precision; bn-MAD) and how consistent the timing patterns are throughout the performance (consistency; *CV*). Both metrics are based on timing deviation, defined as the difference between the performed onset time and the expected (model) onset time for each note:
*Timing deviation = Performed onset – Model onset*.

The bn-MAD was calculated by first computing each note's timing deviation as a fraction of the beat duration at that point in the music, then taking the average of these proportional errors:

$$bnMAD = \frac{1}{n}\sum_{i=1}^{n}\left|\frac{Performed\ onset_i - Model\ onset_i}{Beat\ duration_i}\right| \tag{2}$$

where $n$ is the number of matched notes and *Beat duration$_i$* is the temporal length of one beat in seconds at note $i$'s position (e.g., 1 second at 60 beats per minute; BPM, 0.5 seconds at 120 BPM). This normalization expresses timing deviations as fractions of a beat, making them comparable across different playing tempi. Lower bn-MAD values indicate more precise timing. The beat-normalized nature of this metric means that a bn-MAD of 0.1 represents the same degree of rhythmic imprecision across all tempos—namely, notes are on average 10% of a beat early or late.

To assess timing consistency, we calculated coefficient of variation (*CV*):

$$CV = \frac{Standard\ deviation\ of\ absolute\ timing\ deviations}{Mean\ of\ absolute\ timing\ deviations} \times 100 \tag{3}$$

This captures the relative variability—a performer might have small average deviations but high variability (high *CV*), or large but consistent deviations (low *CV*). Lower *CV* values indicate more stable motor execution.

We calculated rhythm metrics only for notes matched between performance and score (including pitch errors but excluding missing/extra notes) to separate rhythmic accuracy from pitch accuracy. These three metrics were analyzed separately in the subsequent statistical modeling rather than aggregated because they capture theoretically distinct aspects of sight-reading: pitch error rate reflects note identification and visual-motor mapping, bn-MAD captures temporal alignment and pulse maintenance, while *CV* measures motor execution stability. This separation allows examination of how cognitive factors like syntax processing might differentially support these distinct components of sight-reading proficiency. The processed sight-reading performance data are available in the OSF repository (https://osf.io/gaz2t/).

**Self-paced reading task.** For the self-paced reading task, the primary dependent variable was the RTs for each chord. Prior to analysis, several preprocessing steps were applied to ensure data quality. First, only trials where participants correctly answered the chord judgment question were included, ensuring that the RTs reflected engaged processing of the musical material rather than superficial scanning. The raw RTs for these correctly answered trials were then log-transformed to normalize their distribution, as is standard practice in chronometric research where response times typically show positive skew. Finally, outliers were identified and removed by excluding any log-transformed RTs falling beyond ±3 standard deviations from each participant's mean, a procedure that accounts for individual differences in baseline reading speed while removing extreme values that likely reflect lapses in attention or other non-syntactic processing factors. The raw trial-by-trial reading time data for each participant are available in the OSF repository (https://osf.io/gaz2t/).

**Statistical modeling.** We employed linear mixed-effects models using the lme4 package (version 1.1–35.5; Bates et al. [96]) in R (version 4.4.2; R Core Team [97]), with additional functionality from lmerTest (version 3.1–3; Kuznetsova et al. [98]) for significance testing and MuMIn (version 1.48.11; Bartoń [99]) for calculating R-squared values. The significance threshold was set at $\alpha = 0.05$ for all statistical tests.

For Research Question 1, examining whether harmonic syntax can be processed through visual notation alone, we modeled log-transformed RTs as a function of harmonic IC. The model included by-participant random intercepts and slopes for harmonic IC to account for individual differences in both baseline reading speed and sensitivity to harmonic predictability. Initial model diagnostics revealed significant temporal autocorrelation in the residuals (ACF at lag-1 = .18). Temporal autocorrelation occurs when observations close together in time are more similar to each other than observations farther apart—in this case, meaning that a participant's reading time for one chord is partially predictable from their reading time for the immediately preceding chord. This is a common issue in self-paced reading data, where factors such as fatigues, attention fluctuations, or strategic pacing cause

consecutive reading times within a trial to be correlated, potentially violating the independence assumption of regression models. We tested several approaches to address this issue, including adding nested random effects for progression and chord position (1–8) within progressions to account for hierarchical item dependencies, but this failed to reduce autocorrelation (see S1 File for model comparison details). To address this violation of independence assumptions, we included the lag-1 RT as an additional fixed effect predictor. Specifically, for each chord at position $n$, the lag-1 RT variable contains the log-transformed reading time of the immediately preceding chord at position $n$-1. For example, if a participant's log-transformed reading times for chords 1–4 in a progression were 3.30, 3.54, 3.45, and 3.62, the lag-1 RT values for chords 2, 3, and 4 would be 3.30, 3.54, and 3.45, respectively. Because chord 1 (the first chord of each progression) has no preceding observation, it is excluded from the analysis. This approach substantially reduced the autocorrelation to −.093 while maintaining model stability and convergence. Finally, we argue that a random effect of progression number is not necessary on the basis that IDyOM already models local context for each individual chord, taking into account the previous chords that have occurred in a given progression. Thus, the final model structure was: *logRT ~ lag1_RT + harmonic_IC + (1 + harmonic_IC | participant)*.

For Research Question 2, investigating whether sight-reading proficiency moderates the relationship between harmonic IC and RTs—specifically, whether better sight-readers show greater sensitivity to harmonic predictability (steeper slopes, with larger RT increases for less predictable chords, i.e., those with high harmonic IC values)—we extended the base model to include two-way interactions between harmonic IC and each sight-reading performance metric. Separate models were fitted for each combination of sight-reading performance metric (pitch error rate, rhythm precision as measured by bn-MAD, and rhythm consistency as measured by *CV*) and sight-reading condition (tonal and atonal), resulting in six primary models. All continuous predictors were mean-centered (indicated by the suffix "_c" in the variable names) prior to analysis to reduce multicollinearity and facilitate interpretation of main effects in the presence of interactions. The model structure for these analyses was: *logRT ~ lag1_RT_c + harmonic_IC_c × performance_metric_c + (1 + harmonic_IC_c | participant)*.

To explore whether audiation ability further moderated these relationships, we additionally tested three-way interaction models that included NESI scores: *logRT ~ lag1_RT_c + harmonic_IC_c × performance_metric_c × NESI_c + (1 + harmonic_IC_c | participant)*.

Model parameters were estimated using restricted maximum likelihood (REML), and *p*-values for fixed effects were obtained using Satterthwaite's approximation for degrees of freedom. Model comparisons between nested models (two-way vs. three-way interactions) were conducted using likelihood ratio tests, with models refitted using maximum likelihood for these comparisons. Details of the model selection procedure, including the systematic evaluation of alternative specifications to address autocorrelation and convergence handling, are provided in the S1 File. All main analysis scripts used in this study, including model selection and robustness checks, are available in the OSF repository (https://osf.io/gaz2t/).

## Results

### Descriptive statistics

**Sight-reading task.** Table 2 presents descriptive statistics for the three sight-reading performance metrics across tonal and atonal sight-reading conditions. Because each participant performed three pieces per condition, we first calculated each participant's mean score across the three pieces within each condition (tonal and atonal), yielding one aggregated value per participant per condition. These participant-level means were used for all subsequent analyses, including descriptive statistics and paired comparisons. We examined the distributional properties of each performance metric. Shapiro-Wilk tests revealed significant departures from normality for pitch error rate in the atonal condition ($W = 0.86$, $p = .007$), bn-MAD in both conditions (tonal: $W = 0.87$, $p = .010$; atonal: $W = 0.74$; $p < .001$), and *CV* in the tonal condition ($W = 0.89$, $p = .028$). Given these violations, we used Wilcoxon signed-rank tests for all paired comparisons. Fig 4 provides a visual comparison of the three performance metrics across conditions.

As expected, participants demonstrated markedly better pitch accuracy when sight-reading tonal music compared to atonal music. The mean pitch error rate increased nearly four-fold from the tonal to atonal condition. This difference was substantial, with a large effect size.

**Table 2. Descriptive statistics and normality tests for sight-reading performance metrics.**

| Metric and condition | Mean ± SD | Median (IQR) | Range | Shapiro-Wilk | | Paired comparison | | |
|---|---|---|---|---|---|---|---|---|
| | | | | W | p | V | p | r |
| Pitch error rate | | | | | | 0 | < 0.001 | 0.87 |
| Tonal | 0.07 ± 0.04 | 0.06 (0.04–0.10) | 0.02–0.16 | 0.94 | 0.284 | | | |
| Atonal | 0.25 ± 0.09 | 0.28 (0.18–0.34) | 0.09–0.34 | 0.86 | 0.007 | | | |
| bn-MAD | | | | | | 115 | 0.729 | 0.08 |
| Tonal | 0.10 ± 0.05 | 0.09 (0.07–0.11) | 0.04–0.24 | 0.87 | 0.010 | | | |
| Atonal | 0.12 ± 0.09 | 0.09 (0.07–0.11) | 0.05–0.33 | 0.74 | < 0.001 | | | |
| CV (%) | | | | | | 182 | 0.001 | 0.67 |
| Tonal | 211.66 ± 85.54 | 228.97 (124.07–280.80) | 79.50–326.68 | 0.89 | 0.028 | | | |
| Atonal | 137.62 ± 37.00 | 127.22 (115.08–159.49) | 82.00–222.87 | 0.94 | 0.228 | | | |

Abbreviations: bn-MAD: beat-normalized mean absolute deviation; CV: coefficient of variation; SD: standard deviation; IQR: interquartile range; W: Shapiro-Wilk test statistic; p: *p*-value; V: Wilcoxon signed-rank test statistic (paired comparison); r: effect size for Wilcoxon test (0.1 = small, 0.3 = medium, 0.5 = large).

Pitch error rate ranges from 0–1. Lower values indicate better performance (lower pitch error rate, more precise timing, and more consistent timing) for all metrics. As for the paired comparison, Wilcoxon signed-rank tests used due to violations of normality assumptions. Pitch error rate tested with one-tailed test (tonal < atonal); bn-MAD tested with two-tailed test; *CV* tested with one-tailed test (tonal > atonal).

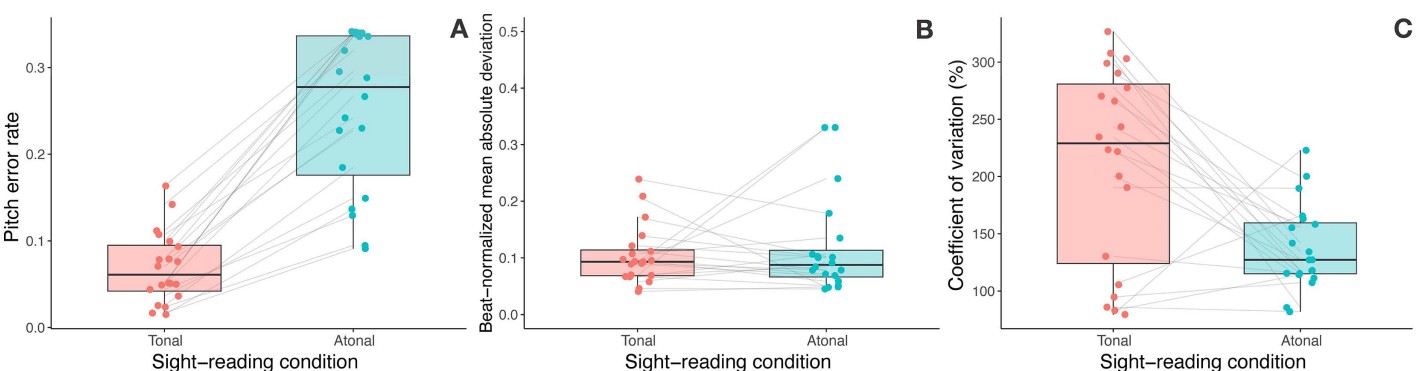

**Fig 4. Comparison of sight-reading performance across tonal and atonal conditions.** *Note.* Box plots show the distribution of (A) pitch error rate, (B) rhythm precision (beat-normalized mean absolute deviation; bn-MAD), and (C) rhythm consistency (coefficient of variation; *CV*) scores. Individual data points are overlaid, with gray lines connecting each participant's performance across conditions to visualize within-subject changes.

For rhythm precision measured by bn-MAD, participants showed comparable performance across conditions, with no statistically significant difference. The effect size was negligible.

Rhythm consistency, measured by *CV*, revealed an unexpected pattern. Contrary to our expectations, participants demonstrated significantly more consistent timing in atonal sight-reading than in tonal sight-reading. This counterintuitive finding may reflect a strategic trade-off: given the substantial difficulty in pitch accuracy for atonal music (as evidenced by the nearly four-fold increase in pitch error rate), participants may have prioritized maintaining steady tempo adherence to the metronome over pitch accuracy. This interpretation aligns with the experimental instruction to continue playing without stopping, suggesting that when faced with challenging pitch content, participants shifted their cognitive resources toward temporal consistency.

**Self-paced reading task.** Each participant completed 46 trials (chord progressions), with each progression containing 8 chords. Because our primary statistical analyses would employ lag-1 RT predictors—which require a preceding

observation—the first chord of each progression cannot be included in the final models. Therefore, to provide descriptive statistics that directly correspond to the data structure used in our linear mixed-effects analyses, we report self-paced reading task performance for chord positions 2–8 only, resulting in 322 observations per participant (7 chords × 46 progressions) and 6,440 total observations across all 20 participants.

Participants demonstrated good engagement with the self-paced reading task, as evidenced by their performance on the chord judgment questions. Accuracy ranged from 67.39% to 97.83%, with a mean of 85.11% ($SD = 9.20$%), well above chance level (50%) while avoiding ceiling effects. This indicates that most participants were actively reading the chord progressions rather than merely advancing through the stimuli, and that the task difficulty was appropriately calibrated.

RTs showed substantial variability both within and between participants (see Fig 5). Before preprocessing, extreme values ranged from 29 ms to 64,908 ms, likely reflecting momentary lapses in attention or technical issues rather than genuine syntactic processing. After removing incorrect trials (14.9%) and values beyond ±3 standard deviations of log-transformed RTs within participants (0.1% of observations), the mean RT per chord was 4,286 ms ($SD = 4,655$ ms), with values ranging from 398 ms to 64,908 ms. The retention of the upper extreme value reflects our within-participant outlier detection approach: this value came from a consistently slow participant ($M = 7,801$ ms, $SD = 8,437$ ms) and fell within their individual ±3 standard deviation range. Specifically, while 64,908 ms appears extreme in raw scale (6.77 standard deviations from this participant's mean), once log-transformed it represents only 2.61 standard deviations, well within our ±3 standard deviation threshold. This approach preserves genuine individual differences while removing only statistically extreme values within each person's reading pattern. At the participant level, mean RTs averaged 4,213 ms ($SD = 2,035$ ms), indicating individual differences in baseline reading speed. For subsequent analyses, RTs were log-transformed to normalize the distribution ($M = 1.08$, $SD = 0.83$).

Reading consistency, measured by the coefficient of variation ($CV$), further highlighted individual differences in processing stability. For raw RTs, the mean $CV$ was 0.82 ($SD = 0.23$), with values ranging from 0.40 to 1.17, while for log-transformed RTs it was 0.73 ($SD = 0.24$), with values ranging from 0.38 to 1.19. These values indicate that some participants maintained relatively consistent reading speeds throughout the task while others showed considerable trial-to-trial variability.

**Audiation task.** Audiation ability, as measured by the NESI test, showed a mean score of 79.59 ($SD = 16.57$) on a 100-point scale. Scores ranged from 33.30 to 91.70, indicating substantial individual differences in the ability to internally represent musical sounds from notation. The distribution contained only seven unique values (33.30, 50.00, 58.30, 66.70, 75.00, 83.30, 91.70) among the 20 participants, reflecting the discrete scoring system of the 12-item test.

## Linear mixed model analyses

### Research question 1: Visual processing of harmonic syntax. Main effects of harmonic IC on reading time

Following the preprocessing procedures outlined in the Methods, the final dataset for linear mixed-effects modeling comprised 5,474 observations. Starting from 7,360 total observations (46 progressions × 8 chords × 20 participants), we first created lag-1 RT predictors to control for temporal dependencies, which necessarily excluded 920 first-position observations (one per progression per participant), leaving 6,440 observations from chord positions 2–8. From these, we excluded 959 trials (14.9%) with incorrect judgment responses and 7 outliers (0.1%) beyond ±3 standard deviations within each participant's log-transformed RTs. This processing order—removing chord 1 before filtering—ensures that outlier detection is based on the actual analyzed data (chords 2–8) rather than including the structurally different first chord position. The final dataset represented 85.0% of chord positions 2–8, confirming that the self-paced reading task successfully captured engaged and consistent processing of chord progressions.

To examine whether harmonic syntax can be processed through visual notation alone, we tested the effect of harmonic IC on RTs in the self-paced reading task, controlling for temporal dependencies through the lag-1 RT predictor ($b = 0.24$, $SE = 0.01$, $t(5354) = 18.28$, $p < .001$, 95% CI [0.22, 0.27]). The model revealed a significant positive

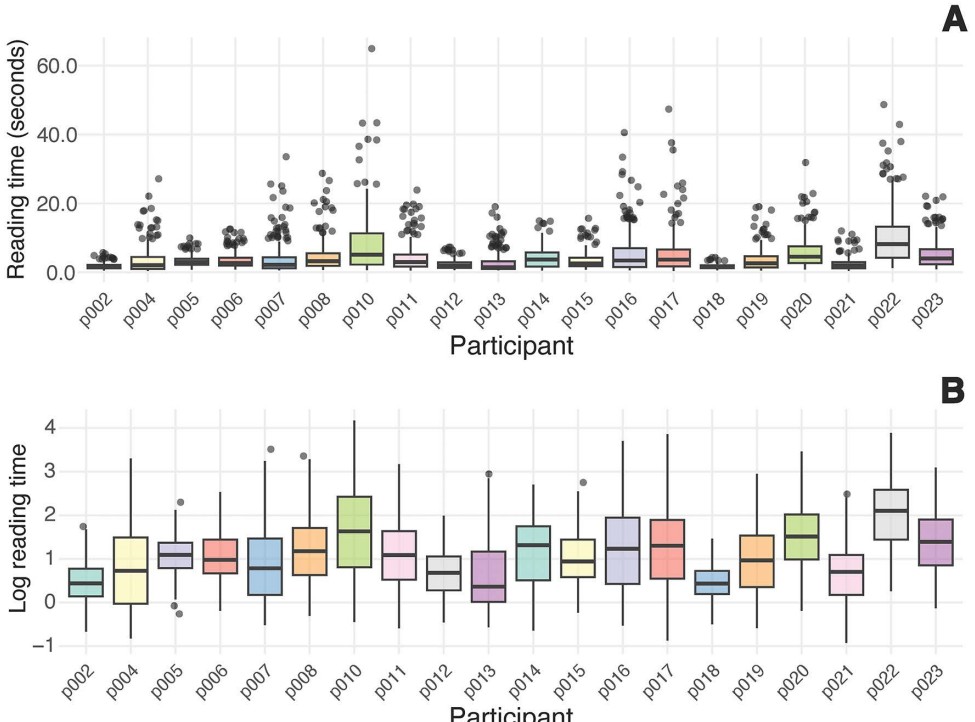

**Fig 5. Individual differences in reading time (RT) distributions for the self-paced reading task.** *Note.* Box plots show the distribution of (A) raw RTs in seconds and (B) log-transformed RTs for each participant.

relationship between harmonic IC and log-transformed RTs ($b = 1.75$, $SE = 0.31$, $t(21.5) = 5.58$, $p < .001$, 95% CI [1.13, 2.36]; see Fig 6).

To contextualize this effect, chords in the third quartile of harmonic IC ($Q_3 = 0.35$) required 6.0% longer RTs than those in the first quartile ($Q_1 = 0.32$), corresponding to approximately 258 ms difference based on the average RT of 4,286 ms. For context, if music is being played at 120 beats/minute (500ms/beat, a common tempo), 258ms accounts for more than half a beat. This processing cost for harmonically less predictable chords provides strong evidence that musicians actively process harmonic syntax during silent music reading, even without auditory input.

The model showed marginal $R^2 = .076$ (fixed effects only) and conditional $R^2 = .208$ (including random effects). The random effects structure revealed reliable individual differences in both baseline reading speed ($SD = 0.28$) and sensitivity to harmonic IC ($SD = 0.09$), with a moderate positive correlation between these parameters ($r = .46$).

## Robustness of the harmonic IC effect on reading time

The finding that harmonically less predictable chords elicited longer reading times could, in principle, be attributable to factors others than syntactic processing. Two alternative explanations warrant consideration. First, chords with higher harmonic IC might also be more visually complex—for instance, containing more accidentals (e.g., sharps, flats, or naturals) or spanning a wider range on the staff—which could increase reading times due to greater perceptual demands rather than sensitivity to harmonic predictability. Second, if high-IC chords happened to be concentrated at the final position of each progression, the observed harmonic IC effect might reflect wrap-up effects: the well-documented tendency for readers to slow down at phrase boundaries to integrate accumulated information, independent of the difficulty of the final item itself [88]. We conducted a series of analyses to evaluate these possibilities.

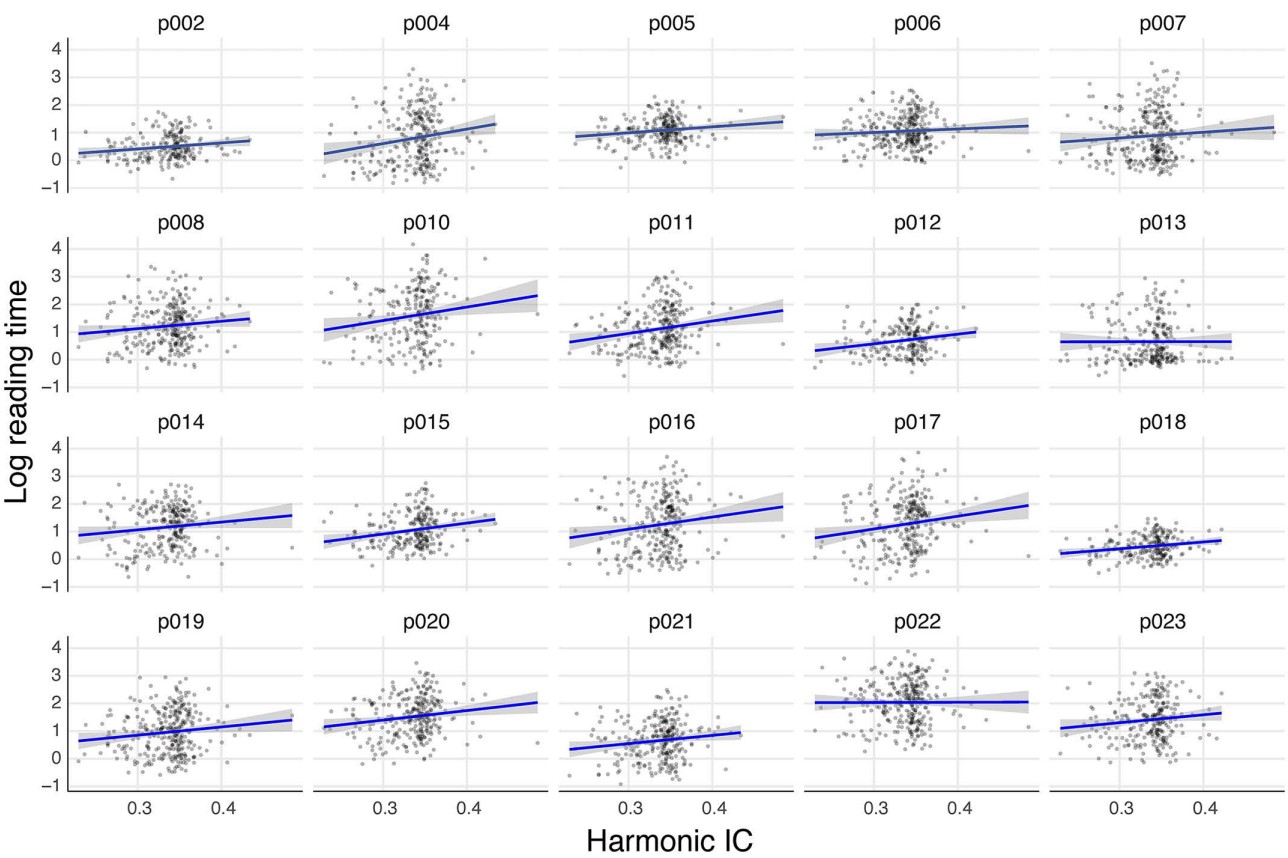

**Fig 6. Individual participant slopes for the effect of harmonic information content (IC) on log-transformed reading time.** *Note.* Each panel shows data from one participant with individual observations (gray dots), linear regression fit (blue line), and 95% confidence interval (shaded area). All participants demonstrated positive slopes, indicating consistent processing costs for harmonically less predictable (high harmonic IC) chords.

To ensure consistency between our exploratory correlation analyses and the linear mixed-effects models, all stimulus-level analyses reported in this section were conducted on chord positions 2–8 ($n = 322$ chords; 46 progressions × 7 positions). Position 1 was excluded because, as mentioned earlier, the linear mixed-effects models included a lag-1 reading time predictor to control for temporal autocorrelation, which necessarily removes the first chord of each progression from the analysis. By applying the same constraint to our correlation analyses, we ensured that the stimulus characteristics examined here directly correspond to the data included in the confirmatory models.

For visual complexity, we examined two indices of visual complexity: accidental count (the number of sharps, flats, or naturals per chord; range: 0–3) and chord spacing (the intervallic distance from the bass to soprano voice; range: 7–40 semitones). The majority of chords contained no accidentals (225 of 322; 69.9%). Neither measure was significantly correlated with harmonic IC (accidentals: Spearman rho $= .02$, $p = .731$; spacing: Spearman rho $= .08$, $p = .166$; $n = 322$), suggesting no significant association between these visual complexity indices and harmonic predictability in our stimulus set. To directly test whether visual complexity could account for the harmonic IC effect, we added accidental count as a covariate in the linear mixed-effects model. The harmonic IC effect remained significant ($b = 1.51$, $SE = 0.31$, $t(21.69) = 4.86$, $p < .001$), with the coefficient decreasing by 13.7% relative to the original model. Similarly, adding chord spacing as a covariate did not substantively alter the results ($b = 1.72$, $SE = 0.32$, $t(23.50) = 5.44$, $p < .001$; 1.7% decrease). As a more stringent test, we restricted the analysis to chords containing no accidentals ($n = 3,874$ observations; 70.8%

of the data), thereby holding visual complexity constant. The harmonic IC effect persisted ($b = 1.10$, $SE = 0.36$, $t(284.4) = 3.06$, $p = .002$). These results are consistent with the interpretation that the relationship between harmonic IC and reading times reflects sensitivity to harmonic predictability, and they suggest that it is unlikely to be fully explained by visual decoding demands.

To evaluate whether wrap-up effects might explain our findings, we examined harmonic IC values across chord positions. Contrary to what would be expected if wrap-up effects were driving the IC-RT relationship, position 8 (the final chord of each progression) had the lowest mean harmonic IC ($M = 0.32$, $SD = 0.03$) among positions 2–8, significantly lower than the average of positions 2–7 ($M = 0.34$, $SD = 0.03$; Mann-Whitney $U = 4510$, $p = .002$, $r = .18$). This pattern suggests that any wrap-up-related slowing at progression endings would, if anything, work against observation and IC effect, making our estimate conservative. Nevertheless, we conducted a sensitivity analysis excluding all final-position chords ($n = 4,692$ observations). The harmonic IC effect remained significant ($b = 1.18$, $SE = 0.35$, $t(320.9) = 3.40$, $p < .001$), providing evidence against the possibility that the IC-RT relationship is primarily an artifact of end-of-sequence processing. Full details of all robustness analyses are provided in S1 File.

**Research question 2: Sight-reading proficiency and syntax processing.** We examined whether individual differences in sight-reading proficiency moderated the relationship between harmonic IC and RTs. In all models reported below, the dependent variable is log-transformed reading time. Table 3 presents the comprehensive results for all three performance metrics across both tonal and atonal sight-reading conditions.

**Interaction effects on reading time**

We did not observe significant moderation from any of the sight-reading performance metrics on the effect of harmonic IC on RTs. The IC × performance metric interactions were not significant across any of the sight-reading conditions tested (all $p$s > .321). The lack of significant interactions suggests that sensitivity to harmonic predictability (i.e., the degree to which reading times increased for less predictable chords) did not vary systematically with sight-reading proficiency in our sample. However, these null findings should be interpreted with caution, as the statistical power to detect such interactions with our sample size may have been limited.

**Main effects of performance metrics on reading time**

Despite the absence of interactions, several performance metrics showed significant main effects on overall RT (Fig 7).

Participants who made more pitch errors during the sight-reading task showed generally slower RTs in the self-paced reading task, with this effect being significant in the tonal condition ($b = 3.44$, $p = .033$), while a similar positive trend was observed in the atonal condition, it did not reach statistical significance ($b = 1.39$, $p = .060$). The standardized effect sizes, as expressed as percentage changes per standard deviation, showed that a one standard deviation increase in pitch error rate corresponded to a 15.1% increase in RT in the tonal condition (approximately 647 ms) and 13.5% in the atonal condition (approximately 577 ms based on average RTs of 4,286 ms). The effect of rhythm precision (bn-MAD) showed marked differences between conditions. In the tonal sight-reading condition, poorer rhythm precision was significantly associated with slower chord reading ($p = .018$), with a 16.4% increase in RT per standard deviation (approximately 703 ms based on average RTs of 4,286 ms). However, this effect was not significant in the atonal sight-reading condition ($p = .397$). The pattern for rhythm consistency ($CV$) was reversed. A significant effect was not observed in the tonal sight-reading condition ($p = .775$), but in the atonal sight-reading condition, participants with more variable timing showed significantly slower chord reading ($p = .039$), with a 14.1% increase per standard deviation (approximately 606 ms based on average RTs of 4,286 ms).

These main effects revealed intriguing condition-specific patterns. While pitch error rate showed similar effects across both sight-reading conditions (tonal and atonal), rhythm metrics showed divergent patterns: bn-MAD only in tonal sight-reading affected RTs, whereas $CV$ only in atonal sight-reading affected RTs. However, without direct statistical comparisons between conditions, we cannot confirm that these condition-specific patterns represent significant differences. The observed patterns may reflect different cognitive resources or strategies employed during tonal versus atonal sight-reading that carry over to silent music-reading tasks.

**Table 3. Linear Mixed model results predicting log-transformed reading times: Harmonic IC × sight-reading performance metrics.**

| Metric and predictor | Tonal condition | | | Atonal condition | | |
|---|---|---|---|---|---|---|
| | b | SE | p | b | SE | p |
| Pitch error rate | | | | | | |
| Intercept | 1.07 | 0.06 | < 0.001 | 1.06 | 0.06 | < 0.001 |
| Lag-1 RT | 0.24 | 0.01 | < 0.001 | 0.24 | 0.01 | < 0.001 |
| Harmonic IC | 1.74 | 0.32 | < 0.001 | 1.76 | 0.31 | < 0.001 |
| Pitch error rate | 3.44 | 1.49 | 0.033 | 1.39 | 0.69 | 0.060 |
| Harmonic IC × Pitch error rate | −3.07 | 7.68 | 0.694 | 3.28 | 3.38 | 0.333 |
| bn-MAD | | | | | | |
| Intercept | 1.07 | 0.06 | < 0.001 | 1.07 | 0.07 | < 0.001 |
| Lag-1 RT | 0.24 | 0.01 | < 0.001 | 0.24 | 0.01 | < 0.001 |
| Harmonic IC | 1.75 | 0.32 | < 0.001 | 1.75 | 0.32 | < 0.001 |
| bn-MAD | 2.98 | 1.15 | 0.018 | 0.68 | 0.79 | 0.397 |
| Harmonic IC × bn-MAD | 2.60 | 6.10 | 0.675 | 2.84 | 3.57 | 0.438 |
| CV | | | | | | |
| Intercept | 1.07 | 0.07 | < 0.001 | 1.07 | 0.06 | < 0.001 |
| Lag-1 RT | 0.24 | 0.01 | < 0.001 | 0.24 | 0.01 | < 0.001 |
| Harmonic IC | 1.75 | 0.32 | < 0.001 | 1.75 | 0.31 | < 0.001 |
| CV | 0.00 | 0.00 | 0.775 | 0.00 | 0.00 | 0.039 |
| Harmonic IC × CV | 0.00 | 0.00 | 0.570 | 0.01 | 0.01 | 0.321 |

Abbreviations: bn-MAD: beat-normalized mean absolute deviation; CV: coefficient of variation; Harmonic IC: harmonic information content; Lag-1 RT: reading time of the previous trial (included as a predictor in the linear mixed model); b: regression coefficient; SE: standard error; p: *p*-value.

The model for bn-MAD in the tonal sight-reading condition produced a minor convergence warning (max|grad| = 0.00305, slightly exceeding the 0.002 threshold), which was resolved by using an alternative optimizer ("bobyqa") available in lme4.

The model for *CV* in the atonal sight-reading condition showed a singular fit due to perfect negative correlation (−1.00) between random intercepts and slopes, indicating insufficient between-participant variation in harmonic IC effects, which was resolved by using a random intercept-only structure (see S1 File for technical details).

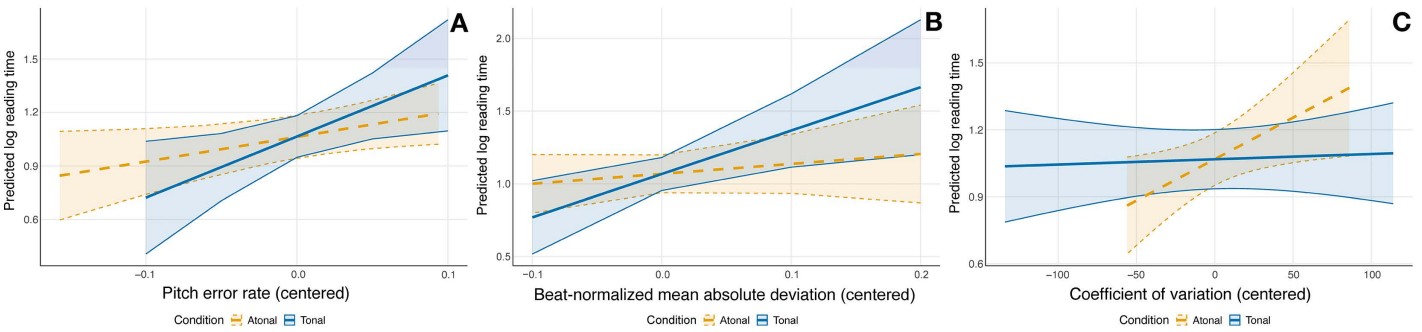

**Fig 7. Model-predicted main effects of sight-reading metrics on log-transformed reading time from the self-paced reading task.** *Note.* (A) Pitch error rate effects, (B) beat-normalized mean absolute deviation (bn-MAD) effects, and (C) coefficient of variation (*CV*) effects. Lines represent predictions from linear mixed models with 95% confidence intervals (shaded areas).

**Three-way interactions with audiation ability.** To explore whether audiation ability might modulate the relationship between sight-reading proficiency and harmonic syntax processing, we tested three-way interactions among harmonic IC, each performance metric, and NESI scores. Table 4 summarizes these analyses.

The model for *CV* in the tonal sight-reading condition also showed a singular fit due to perfect negative correlation (−1.00) between random intercepts and slopes, which was resolved by using a random intercept-only structure (see S1 File for technical details).

We did not find evidence for any of the hypothesized three-way interactions (all *ps* > .108), and likelihood ratio tests consistently favored the simpler two-way models over the more complex three-way models. These results do not provide evidence that audiation ability enhanced or compensated for the relationship between sight-reading proficiency and harmonic syntax processing in our sample. We stress that these null findings should also be interpreted with caution, as any potential effects may have been too small to detect.

## Discussion

This study examined two fundamental questions about harmonic syntax processing in skilled pianists: whether harmonic syntax can be processed through visual notation alone, and how this visual syntax processing relates to sight-reading proficiency. Our findings provide evidence for the first question. Pianists consistently showed longer processing times when reading harmonically less predictable chords, suggesting that harmonic syntax may be processed through symbolic notation in the purely visual modality. However, our second question yielded unexpected results. We did not find a direct relationship between visual syntax processing and participants' sight-reading proficiency or audiation abilities as measured by NESI. Instead, we found that various aspects of sight-reading accuracy were associated with overall reading time of the chord progressions in the self-paced reading task. These findings offer additional evidence for the modality-independence of syntactic processing while illuminating the multifaceted nature of sight-reading as a complex skill. In this discussion, we explore the implications of these two key findings, address theoretical contributions and limitations, and outline directions for future research.

### Evidence of visual syntax processing of harmonic syntax

A central finding of this study is that reading times for visually presented chords appear to vary systematically with harmonic predictability, even in the absence of auditory input. Participants spent on average 6.0% more time (approximately

**Table 4. Three-way interaction results: Harmonic IC × sight-reading performance metrics × NESI score.**

| Performance metric condition | Three-way interaction | | | Model comparison | |
|---|---|---|---|---|---|
| | b | SE | p | LR χ² | p |
| Pitch error rate | | | | | |
| Tonal | −0.20 | 0.77 | 0.803 | χ²(4) = 3.70 | 0.448 |
| Atonal | −0.11 | 0.30 | 0.718 | χ²(4) = 4.39 | 0.356 |
| bn-MAD | | | | | |
| Tonal | −0.16 | 0.31 | 0.607 | χ²(4) = 2.24 | 0.691 |
| Atonal | 0.78 | 0.60 | 0.213 | χ²(3) = 5.73 | 0.125 |
| CV | | | | | |
| Tonal | 0.00 | 0.00 | 0.108 | χ²(2) = 7.10 | 0.029 |
| Atonal | 0.00 | 0.00 | 0.726 | χ²(6) = 0.63 | 0.996 |

Abbreviations: bn-MAD: beat-normalized mean absolute deviation; CV: coefficient of variation; Harmonic IC: harmonic information content; NESI: notation-evoked sound imagery test; b: regression coefficient; SE: standard error; p: *p*-value; LR: likelihood ratio test comparing models with and without three-way interaction terms.

The model for bn-MAD in the atonal sight-reading condition showed a singular fit due to perfect correlation (−1.00) between random intercepts and slopes, which was addressed by removing the correlation between random effects (|| notation), a common approach for resolving singular fits.

258 ms) reading harmonically less predictable chords (high harmonic IC) compared to more predictable ones (low harmonic IC). This effect appeared consistently across all 20 participants, demonstrating the robustness of the phenomenon.

This finding speaks to the *modality gap* we identified in the introduction—the problem that harmonic syntax processing has been explored almost exclusively through the auditory modality. Our results suggest that harmonic syntax processing may be a more abstract, amodal cognitive process not bound to specific sensory modalities. Until now, evidence for harmonic syntax processing has come primarily from ERP responses to auditory stimuli, particularly the ERAN and P600 components ([100] for a review on syntax-related ERP components in music). The few prior studies using visual stimuli did not directly address the issue due to methodological confounds such as the use of familiar melodies [74], concurrent performance requirements [75], or simultaneous audiovisual presentation [72,73], preventing clear measurement of purely visual syntax processing. Importantly, as discussed in the introduction, studies have demonstrated that harmonic syntax can be processed through the visuomotor system during action observation and motor imitation [76,77]—what Sammler et al. [76] termed *embodied* syntax processing. Our study complements this line of work by examining whether similar sensitivity to harmonic predictability can emerge during the processing of symbolic musical notation alone, without explicit motor engagement—a context that could be more directly analogous to the initial stages of sight-reading, and parallel to word reading in language.

A separate methodological concern relates to whether the observed reading time variations might reflect visual decoding demands rather than harmonic processing per se. Our robustness analyses indicated that harmonic IC remained a significant predictor of reading times after controlling for visual complexity as indexed by accidental count and chord spacing in the present study, as well as when the analysis was restricted to chords containing no accidentals. While we cannot rule out all possible sources of visual complexity, this pattern is consistent with the interpretation that our findings may reflect processing of harmonic syntax rather than visual decoding demands alone.

This modality-independent syntax processing parallels findings from language research. Studies have consistently shown that readers fixate longer or slow down when encountering syntactically ambiguous passages (see Clifton et al. [101] for a review), and at the neural level, visually presented grammatical violations elicit P600 responses similar to those in auditory conditions [65,66,68]. Our results suggest that such modality-independence is not unique to language but may reflect a more general cognitive principle for processing structured sequential information. These findings appear consistent with the Shared Syntactic Integration Resource Hypothesis (SSIRH) proposed by Patel [102], which posits shared neural resources for syntactic processing across music and language. Our behavioral evidence raises the possibility that such shared resources, if they exist, might operate across sensory modalities as well, though direct neuroimaging evidence would be needed to evaluate this hypothesis more rigorously.

The phenomenon can also be understood through the lens of predictive coding theory [41,103]. Rather than passively processing sensory input, the brain continuously generates predictions and compares them with actual input, working to minimize prediction error. Greater prediction errors consume more cognitive resources, consistent with the increased processing times we observed for high harmonic IC—less predictable chords. Participants likely formed predictions about upcoming chords based on preceding ones, requiring additional processing when predictions were violated. Crucially, these predictive mechanisms operated without actual sound, using only visual symbols, suggesting that predictive coding can function at an abstract level beyond sensory-specific processing.

While the predictive coding framework offers one interpretation of our findings, it is important to consider what our results can and cannot tell us about the nature of harmonic syntax processing, and to clarify the theoretical constructs involved. Our use of IDyOM to quantify harmonic predictability might raise a conceptual question: how does the statistical predictability measured by information-theoretic models relate to the structural importance emphasized in traditional music-theoretic frameworks? Structural importance, as conceptualized in frameworks such as the Generative Theory of Tonal Music (GTTM, [104]) or Schenkerian analysis, refers to a chord's hierarchical position within tonal organization—with tonic and dominant chords occupying privileged positions. Statistical predictability,

as captured by IDyOM, refers to the conditional probability of an event given its preceding context. These represent conceptually distinct dimensions that need not align. We believe this distinction may help clarify our findings. Corpus studies have shown that chords with roots I and V together constitute over two-thirds of all chords in major-mode passages [105]—these chords are indeed central in terms of overall frequency. Yet frequency alone does not determine IC; IC depends critically on the specific preceding context. Sears et al. [106], using IDyOM to model expectations for cadential patterns in Haydn's string quartets, found that terminal events from cadential contexts (e.g., V ◊ I progressions) received significantly lower IC estimates than non-cadential contexts featuring the same tonic harmony. This suggests that the same chord might be highly predictable when approached via a conventional harmonic formula, yet relatively surprising when appearing in an unexpected sequential position. A chord can thus be structurally central (e.g., a tonic at a cadence) yet highly predictable given what precedes it, or structurally peripheral yet statistically surprising in its local context.

This distinction is relevant to interpreting our findings. One might wonder whether the longer reading times we observed for high-IC chords reflect processing difficulty for statistically less expected events, or whether they instead reflect heightened attention to structurally important elements. Several aspects of our data seem more consistent with the statistical predictability account. First, in our stimulus set, progression-final chords—arguably the most structurally salient positions—had the lowest mean harmonic IC values, and the IC effect remained significant when excluding these positions entirely. If participants were simply attending longer to structurally important chords regardless of predictability, we would expect elevated reading times at final positions; instead, the IC-RT relationship persisted independently of position. Second, if longer reading times reflected positive engagement or aesthetic interest in familiar harmonic events, we might expect trained musicians to linger on conventional, predictable progressions—yet the opposite pattern emerged. (We note, however, that less predictable events might themselves be more interesting or attention-grabbing, in which case this "interest" interpretation would align with rather than contradict the IC-based account.) Nonetheless, we acknowledge that behavioral reading time data alone cannot definitively distinguish between these interpretations. It remains possible that some portion of the reading time variation reflects factors other than processing difficulty, such as deliberate attention to unusual harmonic events—though we note that attention to "unusual" events may itself be driven by their low predictability, making this less of a true alternative explanation than a complementary description of the same phenomenon. Future research employing converging methods—such as eye tracking during score reading or ERP measures of harmonic expectancy (e.g., ERAN)—would help clarify the cognitive processes underlying reading time variations in musical contexts.

One might question whether the 6% (258 ms) increase in RT represents a meaningful effect. However, considering this in the context of real-time music processing reveals its significance. For instance, eye-hand span studies report typical values ranging from 700–1500 ms [20,23,24,27,83]. The 258 ms represents a substantial portion of this look-ahead processing time—far from negligible cognitive cost. Likewise, in a piece performed at 120 BPM (a typical performance tempo), beats occur every 500 ms and eighth notes last merely 250 ms, meaning the 258 ms processing cost represents over half a beat's duration or an entire eighth note's length. Moreover, since this reflects additional processing time for a single chord, the cumulative effect across complex harmonic progressions could reach several seconds.

While all participants showed consistent effects of harmonic IC on processing delays, individual differences in effect magnitude merit future investigation. Some participants showed steeper slopes indicating high sensitivity to harmonic IC, while others showed relatively flat slopes. These individual differences may reflect meaningful variation arising from several factors: differences in explicit harmonic theory knowledge, with more extensive music theory training potentially leading to more sophisticated predictive models; repertoire experience, with greater exposure to specific musical styles strengthening predictions about those harmonic conventions; or cognitive flexibility, with some participants adapting more quickly when predictions fail while others persist longer with existing predictions.

 

In summary, our study provides behavioral evidence that harmonic syntax represents an abstract, amodal knowledge system that can be processed through visual input alone via general cognitive mechanisms like predictive coding. This extends the scope of musical-syntax research beyond the auditory domain—a significant theoretical advance.

## Sight-reading proficiency and visual processing efficiency

Our second research question tested whether superior sight-readers would show greater sensitivity to harmonic syntax. The results diverged from expectations. We did not find evidence that any aspect of sight-reading accuracy—pitch error rate, rhythm precision (bn-MAD), or rhythm consistency (*CV*)—moderated the effect of harmonic IC on RTs. All participants showed similar degrees of processing time increase for less predictable chords. Simultaneously, participants who sight-read more accurately showed overall shorter chord reading times in the self-paced reading task, independent of harmonic IC. This combination of findings reveals that sight-reading skill is not a unitary ability but a phenomenon involving multiple intertwined dimensions.

To understand the null interaction results, we need to distinguish between *possession* and *utilization* of syntactic knowledge. Most prior research has captured how harmonic *knowledge* (but still not syntax) is immediately utilized during actual performance. For example, vocalists' sight-singing accuracy decreases when harmonic predictability is low [16], and unexpected chords trigger immediate eye regressions and pupil dilation [44], showing that harmonic knowledge is actively recruited for real-time prediction and motor planning. Meanwhile, the few studies that independently measured harmonic processing abilities and examined their relationship with sight-reading showed that harmonic context or music-theoretical knowledge individually predict sight-reading ability [17,46], though they still did not directly measure processing of harmonic *syntax*—the functional, hierarchical system of rules between chords. Waters et al. [17], for instance, used only a simple task judging relationships between two chords. Our study aimed to capture precisely this—sensitivity to harmonic syntax during pure visual processing. We calculated harmonic predictability for each chord using IDyOM and measured reading time for 8-bar progressions extracted from actual music repertoires. However, this may have introduced limitations. Unlike simple chord pairs used in Waters et al. [17], our rich musical context may have provided sufficient structural cues for all expert pianists, diluting individual differences. More fundamentally, our participants with nearly 20 years of training may have already shared high levels of syntax processing ability through statistical learning—a potential ceiling effect. At the expert level, individual differences in sight-reading may arise not from the basic ability everyone possesses (*possession* of syntactic knowledge) but from the ability to deploy it in real-time (*utilization*). The dynamic adjustment of eye-hand span [Eye-Time Span for 22,23]—the ability to adjust look-ahead processing in real-time based on musical predictability—exemplifies such utilization. Our task may not have captured this utilization aspect, which may explain our failure to detect significant interactions.

This is where the main effects we discovered—shorter overall chord reading times among more accurate sight-readers—become meaningful. We interpret this effect as reflecting differences in more fundamental visual processing efficiency (i.e., speed of decoding musical notation). This aligns with recent research showing close relationships between sight-reading and visual processing abilities. For instance, Fan et al. [79] found that approximately 10% of sight-reading performance variance was explained by visual fluency for notes alone, an independent explanatory power that persisted after controlling for all other factors (i.e., visual-auditory association, visual-motor association, etc.). The individual differences in overall RT we observed may reflect these differences in basic visual fluency—the ability to quickly and accurately recognize musical symbols. This visual efficiency extends beyond simple perceptual speed. fMRI studies of music reading [107] show that when music-reading experts view notes, extensive multimodal networks including auditory, motor, and somatosensory regions activate automatically alongside visual cortex. Importantly, individuals with higher perceptual fluency for notes showed stronger activation of these multimodal networks. Therefore, the faster chord reading among more accurate sight-readers in our study likely reflects the efficiency of neural circuits that automatically convert visual information into multimodal representations. Furthermore, Arthur et al. [81] demonstrated that this visual advantage is not

music-specific. Sight-reading experts also excelled in working memory capacity and rapid naming of general words, suggesting that sight-reading expertise relates to domain-general visual processing efficiency. The differences in overall RT we observed, independent of harmonic IC, may reflect such general visual processing efficiency.

Taken together, while we did not find the expected interaction between harmonic syntax processing and sight-reading ability, the relationship between sight-reading accuracy and overall RT emerged clearly. This appears to stem from differences in fundamental speed of visual notation processing rather than depth of harmonic syntactic knowledge. These findings align with Kopiez and Lee's [12,108] identification of "speed of information processing" as a key predictor of sight-reading ability in their regression models, and parallel phenomena in language reading where skilled readers maintain consistently fast reading speeds relatively insensitive to text characteristics such as word length or frequency [109,110]. Such cross-domain similarities suggest that skilled reading—whether musical or linguistic—rests on cognitive fluency in efficiently processing visual information beyond sophistication of specific knowledge. Our findings reaffirm the central role of visual processing in the complex skill of sight-reading while showing that individual differences in visual efficiency persist even among experts and can translate into performance differences.

## The role of audiation

We did not find support for the hypothesis that audiation ability would moderate the relationship between visual syntax processing and sight-reading proficiency. NESI scores measuring audiation ability did not show a significant effect on participants' sensitivity to harmonic predictability. This suggests two possibilities. First, the visual harmonic syntax processing we measured may be a cognitive process occurring at a more abstract, symbolic level that does not necessarily involve internal auditory imagery. Second, the audiation abilities of our expert participants may have already been sufficiently high and homogeneous ($M = 79.59$) to show meaningful moderating effects as an individual difference variable.

However, this lack of significant moderating effects on syntax sensitivity should not be interpreted as audiation being unimportant for sight-reading. Multiple prior studies have consistently shown that visual-auditory association abilities are key predictors of sight-reading performance [79,108,111,112]. This suggests that audiation ability may contribute to sight-reading performance through pathways separate from sensitivity to visual harmonic syntax. While visual syntax processing provides abstract, structural predictions about upcoming harmonies (or whatever musical elements that are syntactic), audiation ability may provide concrete auditory templates for sounds to be performed, helping performers monitor their performance in real-time, correct errors, and plan musical expression. Therefore, visual syntax processing and audiation are likely independent, complementary subskills constituting the multi-component skill of sight-reading. Rather than moderating each other, these two abilities may contribute to overall sight-reading performance in parallel, each in their own way. Future research should explore how these independent components—i.e., visual syntax processing, general processing efficiency, and audiation ability—are dynamically integrated and coordinated in actual performance situations.

## Limitations and future directions

While this study explored the novel domain of visual harmonic syntax processing, several follow-up investigations are needed to fully understand the implications and generalizability of our findings. First is the homogeneity of our participant sample. Our participants were highly skilled pianists with extensive training. While this homogeneity enhanced internal validity, it limits generalizability. The relationship between syntactic processing and sight-reading ability may differ across expertise levels. Studies including broader expertise spectrums from beginners to experts might reveal stronger correlations at stages where syntactic knowledge is not yet fully internalized. The main effect we discovered—that more accurate sight-readers read chords faster overall—also warrants verification for universality. If this relationship appears consistently from beginners to experts, it would suggest that visual fluency as a universal foundation for sight-reading development. If it appears only at specific developmental stages, it would suggest that qualitative transition points in sight-reading

expertise. Understanding these developmental trajectories could provide direct implications for designing effective sight-reading instruction timing and methods.

Second, the interpretation of reading times in musical contexts may involve complexities not fully addressed in the present study. While self-paced reading time modulation by predictability is commonly taken as evidence for on-line processing in psycholinguistic research, we acknowledge that reading times in musical notation reading could also reflect factors such as attention to structurally salient elements, aesthetic engagement, or visual decoding demands that may not be entirely separable from syntactic processing with behavioral measures alone. Converging evidence from other methodologies would therefore strengthen conclusions about the nature of visual harmonic syntax processing. Eye-tracking could reveal specific visual search strategies when encountering less predictable chords—including fixation durations, attention allocation patterns, and regression to previous chords—which might help clarify what cognitive processes our measured reading time differences actually reflect. Furthermore, ERP studies could examine whether visually presented harmonic violations elicit neural responses similar to ERAN or P600 observed in auditory conditions, which would provide more direct evidence regarding the potential modality-independence of syntactic processing. Using fMRI or MEG to investigate the involvement of auditory cortex during visual syntax processing could also help elucidate the role of audiation. If visual syntax processing can occur without substantial auditory cortex activation, this would be consistent with the possibility of a more abstract syntactic processing pathway, though such conclusions would require careful interpretation

## Conclusions

This study provides evidence suggesting that sensitivity to harmonic predictability may extend to the visual modality: even without auditory input, participants showed longer reading times for less predictable chords, suggesting that expectation-based harmonic processing may occur through symbolic notation alone. This finding highlights the importance of visual processing in musical syntax research and opens new questions about modality-independent representations: our results suggest that harmonic-syntactic knowledge may be accessible not only through auditory perception and visuomotor simulation, but also through the processing of abstract visual representations in written notation. However, this visual syntax processing ability itself did not appear to be a significant predictor of sight-reading proficiency differences among our sample of highly skilled pianists. Instead, sight-reading proficiency appeared to be more closely related to general visual processing efficiency for decoding musical notation. These findings suggest that sight-reading expertise may not be reducible to a single syntax processing ability but rather emerges from the coordination of multiple components including visual processing fluency, syntactic knowledge, and audiation ability. This study sought to isolate these components of this complex skill, providing evidence consistent with the interpretation that while visual syntax processing may be a basic ability shared by experts, general visual processing efficiency could be another key dimension underlying individual differences. These findings may contribute to our theoretical understanding of the expert musical mind and could also offer practical implications for music education by highlighting the potential importance of training that enhances visual processing fluency alongside traditional, for example, harmony-oriented instruction. Ultimately, we hope these findings contribute to understanding the multimodal nature of music cognition and exploring the universality and specificity of human capabilities for structured information processing.

## Supporting information

**S1 File. Supplementary methods and results.** The file contains additional methodological details including participant exclusion criteria, atonalization procedure for creating the atonal sight-reading materials, the complete participant questionnaire, detailed statistical analyses addressing temporal autocorrelation in self-paced reading data, comprehensive documentation of how we handled convergence issues in the mixed-effects models, and robustness analyses examining the consistency of our findings across alternative analytical approaches.
(DOCX)

 

## Acknowledgments

The authors would like to thank the members of the Music and Mind Lab at the Indiana University Jacobs School of Music for their invaluable feedback and insights during the development of this research. We are also grateful to Thomas Cooke for his assistance in preparing the stimuli, and Richard Zhang and Charles Enyinna Nwakanma for their assistance with manuscript formatting and preparation.

## Author contributions

**Conceptualization:** Yeoeun Lim, Andrew Goldman.

**Data curation:** Yeoeun Lim.

**Formal analysis:** Yeoeun Lim.

**Funding acquisition:** Andrew Goldman.

**Investigation:** Yeoeun Lim.

**Methodology:** Yeoeun Lim, Andrew Goldman.

**Project administration:** Yeoeun Lim, Andrew Goldman.

**Resources:** Andrew Goldman.

**Software:** Yeoeun Lim, Andrew Goldman.

**Supervision:** Andrew Goldman.

**Validation:** Yeoeun Lim.

**Writing – original draft:** Yeoeun Lim, Andrew Goldman.

**Writing – review & editing:** Yeoeun Lim, Andrew Goldman.

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
