## [Decision Letter · Decision Letter 0]

22 Dec 2025

Dear Dr. Goldman,

Thank you for submitting your manuscript to PLOS ONE. After careful consideration, we feel that it has merit but does not fully meet PLOS ONE’s publication criteria as it currently stands. Therefore, we invite you to submit a revised version of the manuscript that addresses the points raised during the review process.

The external reviews (appended below) identify significant strengths in your experimental design but raise substantial concerns regarding your theoretical framing and computational modeling. Additionally, during my own assess<source-footnote _nghost-ng-c3989515971="" ng-version="0.0.0-PLACEHOLDER"></source-footnote>ment of the manuscript, I identified a critical <source-footnote _nghost-ng-c3989515971="" ng-version="0.0.0-PLACEHOLDER"></source-footnote>methodological discrepancy regarding statistical controls for visual complexity that was not explicitly detailed by the reviewers but must be resolved.

<sources-carousel-inline _nghost-ng-c3608714212="" ng-version="0.0.0-PLACEHOLDER"><source-inline-chips _ngcontent-ng-c3608714212="" _nghost-ng-c2631690858="" class="ng-star-inserted"><source-inline-chip _ngcontent-ng-c2631690858="" _nghost-ng-c187302697="" class="ng-star-inserted"></source-inline-chip></source-inline-chips></sources-carousel-inline>

In Figure 1, the penultimate chord (dominant) and the final chord (tonic) are assigned IC values of 0.39 and 0.35, respectively, which are among the highest values in the sequence<source-footnote _nghost-ng-c3989515971="" ng-version="0.0.0-PLACEHOLDER"></source-footnote>. If your high-IC chords are predominantly located at phrase boundaries, the correlation between IC and RT may simply reflect participants lingering at the double-barline. You must demonstrate that your results hold when controlling for chord position (specifically the final position).

I share the serious concerns raised by both reviewers regarding the implementation of IDyOM.

You must address Reviewer 1's observation that the model appears to identify stable cadential chords as "high entropy/unpredictable"<source-footnote _nghost-ng-c3989515971="" ng-version="0.0.0-PLACEHOLDER"></source-footnote>. If the model assigns high information content to structurally stable events, the validity of using these values to predict reading times is fundamentally compromised.

Also, you must clarify how a model typically used for monophonic sequences was applied to polyphonic/chordal data, as noted by Reviewer 2.

I agree with the reviewers that the claim that "it is still unknown whether harmonic syntax can be processed without auditory input, through visual symbols alone" <source-footnote _nghost-ng-c3989515971="" ng-version="0.0.0-PLACEHOLDER"></source-footnote> is overstated. You must revise the Introduction to accurately reflect the existing state of the art (citing the literature noted by Reviewer 2, such as Sammler et al., 2013 and Bianco et al., 2016), extending the field rather than claiming a "first empirical demonstration"

We look forward to receiving your revised manuscript.

Kind regards,

Bruno Alejandro Mesz, Ph.D.

Academic Editor

PLOS One

**Journal Requirements:**

1. When submitting your revision, we need you to address these additional requirements. Please ensure that your manuscript meets PLOS ONE's style requirements, including those for file naming. The PLOS ONE style templates can be found at https://journals.plos.org/plosone/s/file?id=wjVg/PLOSOne_formatting_sample_main_body.pdf and https://journals.plos.org/plosone/s/file?id=ba62/PLOSOne_formatting_sample_title_authors_affiliations.pdf 2. PLOS requires an ORCID iD for the corresponding author in Editorial Manager on papers submitted after December 6th, 2016. Please ensure that you have an ORCID iD and that it is validated in Editorial Manager. To do this, go to ‘Update my Information’ (in the upper left-hand corner of the main menu), and click on the Fetch/Validate link next to the ORCID field. This will take you to the ORCID site and allow you to create a new iD or authenticate a pre-existing iD in Editorial Manager. 3. We note that this data set consists of interview transcripts. Can you please confirm that all participants gave consent for interview transcript to be published? If they DID provide consent for these transcripts to be published, please also confirm that the transcripts do not contain any potentially identifying information (or let us know if the participants consented to having their personal details published and made publicly available). We consider the following details to be identifying information:- Names, nicknames, and initials- Age more specific than round numbers- GPS coordinates, physical addresses, IP addresses, email addresses- Information in small sample sizes (e.g. 40 students from X class in X year at X university)- Specific dates (e.g. visit dates, interview dates)- ID numbers Or, if the participants DID NOT provide consent for these transcripts to be published:- Provide a de-identified version of the data or excerpts of interview responses- Provide information regarding how these transcripts can be accessed by researchers who meet the criteria for access to confidential data, including:a) the grounds for restrictionb) the name of the ethics committee, Institutional Review Board, or third-party organization that is imposing sharing restrictions on the datac) a non-author, institutional point of contact that is able to field data access queries, in the interest of maintaining long-term data accessibility.d) Any relevant data set names, URLs, DOIs, etc. that an independent researcher would need in order to request your minimal data set. For further information on sharing data that contains sensitive participant information, please see: https://journals.plos.org/plosone/s/data-availability#loc-human-research-participant-data-and-other-sensitive-data If there are ethical, legal, or third-party restrictions upon your dataset, you must provide all of the following details (https://journals.plos.org/plosone/s/data-availability#loc-acceptable-data-access-restrictions):a) A complete description of the datasetb) The nature of the restrictions upon the data (ethical, legal, or owned by a third party) and the reasoning behind themc) The full name of the body imposing the restrictions upon your dataset (ethics committee, institution, data access committee, etc)d) If the data are owned by a third party, confirmation of whether the authors received any special privileges in accessing the data that other researchers would not havee) Direct, non-author contact information (preferably email) for the body imposing the restrictions upon the data, to which data access requests can be sent 4. If the reviewer comments include a recommendation to cite specific previously published works, please review and evaluate these publications to determine whether they are relevant and should be cited. There is no requirement to cite these works unless the editor has indicated otherwise. 

Reviewers' comments:

**Comments to the Author**

1. Is the manuscript technically sound, and do the data support the conclusions?

Reviewer #1: No

Reviewer #2: Yes

2. Has the statistical analysis been performed appropriately and rigorously?

Reviewer #1: Yes

Reviewer #2: Yes

3. Have the authors made all data underlying the findings in their manuscript fully available?

Reviewer #1: Yes

Reviewer #2: No

4. Is the manuscript presented in an intelligible fashion and written in standard English?

Reviewer #1: Yes

Reviewer #2: Yes

**Reviewer #1:** The study addresses the processing of harmonic syntax through visual notation, conceptualizing such processing ability as sensitivity to the predictability of chords, and asking whether it is related to sight-reading proficiency. The study addresses harmonic processing in terms of the reading times of chords in a self-paced reading task. In statistical analyses, the reading times are modeled by computationally produced values of harmonic predictability for chords, as well as by participants’ performance in a sight-reading task and audiation task. The authors could not find an effect of sight-reading or audiation skill on the reading times in the self-paced reading task, but they did find a connection between the calculated harmonic predictability of the chords and the reading times.

There are many laudable features in the manuscript. For instance, the research background is well explained, the measures of sight reading proficiency are well chosen, and the statistical analyses are appropriately carried out. The supplemental file gives a lot of useful additional information about the analyses, and the language is very good. However, I find that there are a number of interlinked problems that call the results of the study into question. The problems have to do with the authors’ notion of processing harmonic syntax and how it is operationalized in the study, as well as the unusual conception of harmonic predictability that the main results rely on. I have tried to explain these problems in more detail below. Unfortunately, I think the problems run too deep into the basic study design that they cannot be very easily mended without a total reanalysis of the data, most likely using another approach to the question of harmonic predictability. Therefore, my recommendation is resubmission after a major revision.

In the following, I will first address three important problem areas, and thereafter list a number of smaller suggestions or corrections. I hope these observations could help the authors as they continue to work on their study.

Reading times

The self-paced reading task is central to the whole study, so it should be much better explained. First, it is unclear how exactly the stimuli were presented (did the participants see the chords sequentially at the center of the screen, or were the chords somehow successively highlighted in an entire visible score, or what?). Second, it appears that there was no specific instruction in the self-paced reading task that would make it obvious how the reading times should be interpreted. Now, the authors simply assume that longer reading times indicate “processing costs,” but this seems an a priori assumption, given that the task instructions did not seem to have called for quick responses. Alternatively, longer reading times might also reflect interest, attention to structurally central elements, or the like. Notice that such interpretations would be quite contrary to what is now simply taken for granted. In my view, the interpretation of the main results therefore lies on shaky ground: we cannot actually be sure what the reading time data is about, and thus what is being explained.

This unclarity of the main focus is somewhat masked in the abstract where it is suggested that the study addresses the research gap regarding “whether sensitivity to harmonic syntax predicts better sight-reading” (p. 2). Given the analyses conducted in this study, this is not really the gap that is addressed. In the study, the authors instead focus on questions such as whether various sight-reading metrics moderate the effect of harmonic predictability on reading times in the self-paced reading task. This is clearly a different question, and one that again direct focus to understanding the reading times in the silent-reading task which has been unconstrained by instructions that would make it easily interpretable.

Predictability

Regarding the self-paced reading task, the authors write that ”The distribution of the IC values (Fig 2) confirmed that our stimuli encompassed a representative range of harmonic predictability typical of Western tonal music” (p. 21). This seems questionable. In the example chord progression (Fig. 1), all chords represent triads within the diatonic scale, and the harmonic IC values range between 0.31–0.39. According to Fig. 2, most of the chords in the stimuli indeed fell within this range. In the example progression (Fig. 1), however, the ”most surprising” events are now the beginning tonic triad and the penultimate dominant triad! The authors should remember that these chords are taken as almost axiomatic points of stability in standard theories of Western tonal syntax (e.g., Schenker or GTTM). See, for instance, GTTM’s time-span reductions in which the structural beginning and end of a composition (I […] V–I) are taken as the most stable events (e.g., Lerdahl & Jackendoff, 1983, p. 144).

It is indeed a bit difficult to understand why, for instance, the third last chord (I) of Fig. 1 is evaluated as more predictable as either of the two last cadential chords (V and I). It is hard to think about any usual kind of notion of harmonic predictability that this would conform with. If these are really the values produced by the Information Dynamics of Music framework, the authors should understand that such values are simply not convincing and that readers may well raise questions about the well-foundedness of the premises behind the computational procedure. At the very least, it certainly should not be the case that a ”representative range of harmonic predictability typical of Western tonal music” is one in which chords that are usually taken as the most stable and predictable ones turn out to be the “least predictable” events! The authors should seriously consider using some other computational model for assessing the syntactic role of chords (e.g., Lerdahl’s Tonal Pitch Space model).

The authors also easily slide from their special notion of predictability to talking about functional hierarchies. The reader may ask what grounds there are to claim that harmonic predictability, as operationalized here, has to do with “the functional, hierarchical system of rules between chords” (p. 47)? Again, the authors should really understand how questionable it is to suggest that the main functional chords in a composition as the final V and I in Fig. 1 are here claimed to be “less predictable” chords. It is highly misleading to suggest that such a strange conception represents “the functional, hierarchical system of rules between chords.”

All this is also obviously linked to the interpretation of the results. Now, the authors write that there were “consistent processing costs for harmonically less predictable (high harmonic IC) chords” (p. 37). To use the above example again, the authors appear to believe the cadential dominant triad to be among the “harmonically less predictable” chords, and that the unpredictability of such a chord would lead to higher “processing costs.” However, disregarding the algorithm used here, any basic understanding of tonal syntax should actually suggest such a chord to be most highly predictable in Bach-style chorale writing in the penultimate position. If so, we have reason to revisit the above criticism regarding the assumption that longer reading times indicate processing costs: perhaps the readers might have shown more interest in such structurally important chords?

Processing of harmonic syntax

The authors claim that “it is still unknown whether harmonic syntax can be processed without auditory input, through visual symbols alone” (p. 11). Let me just note first that, at face value, this may sound like a rather absurd statement, given that generations of music students have been taught to do exercises in harmonic analysis by silently reading musical notation! That surely counts as processing harmonic syntax. Therefore, I would strongly recommend that the authors do not use these kinds of generalizing statements when talking about sensitivity to chord predictability. The conclusion “This study provides the first empirical demonstration that harmonic syntax can be processed visually without auditory input” (p. 52) thus appears misleading, and may seem like an attempt to artificially inflate the value of the study.

The authors do explain what they mean with processing harmonic syntax (even though this could be done earlier), and it helps a bit, but not entirely. Visually processing harmonic syntax is explained to mean “the ability to extract harmonic relationships from musical notation without auditory input, measured through sensitivity to harmonic predictability,” “through the degree to which reading times increase according to the harmonic predictability of the chords in the self-paced reading task” (pp. 13–14). In other words, using more silent-reading time for “less predictable chords” than “predictable” ones “indicates more efficient extraction and utilization of syntactic information from visual notation” (p. 14). However, assuming that reading times indeed reflect processing costs, we may now ask why would relative difficulties in reading less predictable chords indicate “more efficient extraction and utilization of syntactic information”? Given that all of the chords seem to have been diatonic triads, why couldn’t an efficient extraction be conversely reflected by smaller differences in reading time between the different chords? Again, it seems that the authors are simply assuming something.

Note that if some of chords evaluated by the algorithm as “less predictable” are actually among the most common tonal chords (see above), the readers’ use of time for reading the different chords could also then be interpreted as an indication of processing harmonic syntax. However, in that case the interpretation from “predictability” would need to be revised, e.g., by suggesting that the participants have used most reading time for the structurally central elements. There seems to be nothing in the study design that would rule out such a contrary interpretation. The participants were experienced musicians: why should their relative use of time between tonal triads primarily be interpreted in terms of relative surprisingness, instead of assuming that their reading process can also reflect something of the relative functional hierarchies functional hierarchies in the tonal system?

Even given the restricted definition of “processing harmonic syntax” in terms of sensitivity to chord predictability, the authors’ broader claims tend to slide into more general conclusions regarding “a more general cognitive principle for processing structured sequential information” (p. 44). However, it should be remembered that the data has to do with reading times for chords. The same conclusions might have been reached by a reductio ad absurdum version of the study, where the stimulus chords would include both very conventional chords (e.g., {C, E, G}) and less conventional and syntactically unclear ones (e.g., {C, Db,Gb}). It could be expected that silent reading times for the latter would often be longer than for the former, and that some similar differences could be obtained in a parallel listening task. I suspect, then, that it would be rather easy to get such a result with stimuli that would also more convingly align with standard notions of chord predictability.

But the question remains if there is anything interesting about such a result (whether in the present study or in my above hypothetical example)? How do such results about reading times help us draw the conclusion “that music and language not only share neural resources for syntactic processing but that these resources operate across sensory modalities” (p. 44)? How should the silent-reading times on the scale of seconds (that may or may not reflect the subjective difficulty of the stimuli, their predictability, or the significance attributed to them) be related to involuntary brain responses in listening studies that appear on a much smaller time scale? It appears to me that the authors are presenting results that are not very surprising, but also strongly overstating their importance for general conclusions about music cognition. In the authors’ approach, the notion that something “can be processed” (p. 46) in a given modality comes very cheap: it does not seem to require much more that we can show that some syntactic elements take more time to tackle than others—whatever the task is and whatever the potential reasons for using different amounts of time.

Minor comments

P. 27: In the formula for CV, should the phrase over the line be “Standard deviation of absolute timing deviations”? Notice that the absence of the word “absolute” may be a bit confusing: “abolute deviations” are mentioned in the denominator, but otherwise, this has not been explained, and the reader still has in mind the beat-related deviations from the previous formula. It might be useful to define “timing deviation” outside of the bnMAD formula so that it can also be understood for CV.

P. 31: If I understand correctly from the paired comparisons using the Wilcoxon signed-rank test, the authors must have first calculated mean participant values for each of the accuracy metrics. This does not seem quite clear now. For instance, the formula for Pitch Error Rate mentions the number of total notes in a score (p. 27), but there were three scores in both of the (tonal and atonal) conditions (p. 17). Please make it clear if means were taken first across the three scores before the Wilcoxon signed-rank tests.

P. 36: I think I understand the basic idea, but I think it would be still good to explain in some more detail how you “created lag-1 RT predictors.” What does it exactly mean? Could you give an example of the values of such a variable?

Pp. 37–38: It should be more clearly stated before and in Table 3 that what is predicted here are reading times.

P. 42/line 792: “These results fail to provide evidence that audiation ability neither enhanced nor compensated…” Because of “fail,” this becomes a double negation which is clearly not meant here.

**Reviewer #2:**  While the manuscript presents a well-designed and carefully executed study, several issues require clarification and revision.

First, I disagree with the authors’ statement of the knowledge gap. The manuscript claims that “it is still unknown whether harmonic syntax can be processed without auditory input, through visual symbols alone” (line 189). However, prior work has already demonstrated that musical syntax can be processed in the absence of auditory input using purely visual stimuli (e.g., Sammler et al., 2013, Cortex; Bianco et al., 2016, NeuroImage). This body of literature should be acknowledged and discussed. Moreover, the authors should clarify the specific contribution of the present study beyond these existing findings.

Second, the application of the monophonic IDyOM model to harmonic material is unclear. IDyOM operates on sequences of the pitches or intervals of individual notes as tokens in an n-gram framework. It is therefore not obvious how information content was computed for homophonic and/or polyphonic materials, including the ABRSM sight-reading pieces and the four-part chords used in the self-paced reading task. For example, was only a single voice (e.g., the highest pitch) analyzed and subsequently interpreted as harmonic information content? The procedure requires explicit clarification to ensure that the reported measures are conceptually and methodologically valid.

Third, the training procedure for IDyOM raises concerns about overfitting. Because IDyOM already incorporates local context via its short-term viewpoint, additional training on the test stimuli themselves appears unnecessary and potentially problematic. If the intention was to fine-tune the model to stylistic properties, training on separate materials drawn from the same corpus would be more appropriate. This concern is particularly relevant given the apparent stylistic similarity between the Bach corpus and the test materials.

Finally, I understand that PLOS One requires data availability for all publications. I believe the (minimal) raw data should already be available for review as well. Otherwise, how would reviewers assess the scientific rigor and integrity of the manuscript? Please include the data (or a link to the data) in the revision.

I would be happy to review a revised version of the manuscript that addresses these points.

**Do you want your identity to be public for this peer review?** For information about this choice, including consent withdrawal, please see our Privacy Policy

Reviewer #1: **Yes:** Erkki Huovinen

Reviewer #2: No

---

## [Author Response · Author response to Decision Letter 1]

6 Feb 2026

We have included a PDF called "Response to Reviewers.pdf" in our resubmission files. Please refer to that document for our response. Thank you.

---

## [Editor Report · Decision Letter 1]

23 Feb 2026

Visual processing of musical syntax and its relationship to sight-reading ability

PONE-D-25-62639R1

Dear Dr. Goldman,

We’re pleased to inform you that your manuscript has been judged scientifically suitable for publication and will be formally accepted for publication once it meets all outstanding technical requirements.

Kind regards,

Bruno Alejandro Mesz, Ph.D.

Academic Editor

PLOS One

Additional Editor Comments (optional):

Thank you for submitting the revised version of your manuscript, "Visual processing of musical syntax and its relationship to sight-reading ability," to PLOS ONE. I am pleased to inform you that your manuscript has been accepted for publication.

The revised manuscript and your detailed response to reviewers demonstrate a rigorous addressal of the concerns raised during the initial review. Congratulations on this contribution to the field of music theory and cognitive science.
---

## [Editor Report · Acceptance letter]

PONE-D-25-62639R1

PLOS One

Dear Dr. Goldman,

I'm pleased to inform you that your manuscript has been deemed suitable for publication in PLOS One. Congratulations! Your manuscript is now being handed over to our production team.

Kind regards,

on behalf of

Dr. Bruno Alejandro Mesz

Academic Editor

PLOS One